# Transcription factor paralogs orchestrate alternative gene regulatory networks by context-dependent cooperation with multiple cofactors

Siqian Feng[1,2], Chaitanya Rastogi [3], Ryan Loker[1,2,4,6], William J. Glassford[1,2], H. Tomas Rube[3,7], Harmen J. Bussemaker [3,5] & Richard S. Mann [1,2,5✉]

In eukaryotes, members of transcription factor families often exhibit similar DNA binding properties in vitro, yet orchestrate paralog-specific gene regulatory networks in vivo. The serially homologous first (T1) and third (T3) thoracic legs of *Drosophila*, which are specified by the Hox proteins Scr and Ubx, respectively, offer a unique opportunity to address this paradox in vivo. Genome-wide analyses using epitope-tagged alleles of both Hox loci in the T1 and T3 leg imaginal discs, the precursors to the adult legs and ventral body regions, show that ~8% of Hox binding is paralog-specific. Binding specificity is mediated by interactions with distinct cofactors in different domains: the Hox cofactor Exd acts in the proximal domain and is necessary for Scr to bind many of its paralog-specific targets, while in the distal leg domain, the homeodomain protein Distal-less (Dll) enhances Scr binding to a different subset of loci. These findings reveal how Hox paralogs, and perhaps paralogs of other transcription factor families, orchestrate alternative downstream gene regulatory networks with the help of multiple, context-specific cofactors.

[1] Department of Biochemistry and Molecular Biophysics, Columbia University, New York, NY, USA. [2] Mortimer B. Zuckerman Mind Brain Behavior Institute, Columbia University, New York, NY, USA. [3] Department of Biological Sciences, Columbia University, New York, NY, USA. [4] Department of Genetics and Development, Columbia University, New York, NY, USA. [5] Department of Systems Biology, Columbia University, New York, NY 10027, USA. [6] Present address: Department of Biology, New York University, New York, NY, USA. [7] Present address: Department of Bioengineering, University of California, Merced, CA, USA. ✉email: rsm10@columbia.edu

Serial homology refers to animal body parts that are recognizably similar to each other, yet have distinct morphological characteristics that are optimized for carrying out specialized functions[1]. The forelimbs and hindlimbs of tetrapod animals and the wings and halteres of dipteran insects are both examples of serially homologous appendages within an organism. The concept of homology is also useful for comparing structures between species, such as the hindlegs of a kangaroo versus the hindlegs of a horse. In these examples, evolutionary forces sculpted morphological differences between these appendages to optimize their functions in each species. For both types of morphological variation, the homeodomain transcription factors encoded by the Hox genes, together with the gene regulatory networks they control, play a central role. To diversify structures within an organism, Hox genes have duplicated, allowing them to alter their activities and expression domains, thus facilitating morphological modifications to appendages and other body parts[2]. Analogously, on an evolutionary time scale, modifications of Hox gene networks have played a central role in generating the vast diversity of animal morphologies in biology that exists today[3,4].

Although changes in Hox gene networks are a major driving force in animal morphological diversity, the underlying mechanisms are not well understood. For instance, for any pair of homologous structures, we are largely ignorant about how many and what types of changes to Hox gene regulatory networks are required to modify morphologies, and how many are directly controlled by Hox transcription factors. Second, as transcription factors, all Hox paralogs have very similar DNA-binding homeodomains and binding specificities, raising the fundamental question of how different Hox paralogs execute distinct, yet related, gene regulatory networks in homologous body parts[5–7]. One answer to this question is that DNA-binding cofactors, in particular Extradenticle (Exd) in Drosophila and Pbx in vertebrates, reveal novel latent DNA-binding specificities upon heterodimerization with Hox factors[8]. However, these cofactors are only available in a subset of Hox-expressing domains, implying that there are additional cofactors and/or non-DNA-binding mechanisms that are used to discriminate between Hox functions in serially homologous structures.

In this study, we address these and related questions in the context of a classic example of serial homology, namely, how two Drosophila Hox proteins—Sex combs reduced (Scr) and Ultrabithorax (Ubx)—achieve their paralog-specific functions to specify distinct leg morphologies in the first (T1) and third (T3) thoracic segments, respectively. Although the transcriptomes of the larval precursors of the legs are very similar, a comparison between the chromatin immunoprecipitation followed by deep sequencing (ChIP-seq) profiles of Scr and Ubx revealed that ~8% of binding by these Hox proteins is paralog-specific, suggesting that the different leg morphologies are initiated at least in part by differences in Hox binding to a small set of enhancers. Further, we show that differential chromatin accessibility or Scr and Ubx monomer binding specificities are not sufficient to account for paralog-specific binding. On the other hand, comparing the ChIP-seq profiles between wild type and a mutated Scr that is unable to heterodimerize with Exd revealed that many, but not all Scr-specific binding events are Exd-dependent. We further identified the homeodomain protein Distal-less (Dll) as a previously unknown Scr cofactor capable of enhancing Scr–DNA binding in cells where Exd is not available. Reporter gene assays support the idea that Dll, as well as additional cofactors, contribute to Scr's specific activities in the T1 leg. Overall, using a combination of whole-genome and mechanistic approaches, we demonstrate that to generate distinct morphologies in serially homologous body parts, Hox proteins depend on multiple, region-specific DNA-binding cofactors to directly modify gene regulatory networks.

## Results

### Paralog-specific Hox expression and function in developing Drosophila legs

In Drosophila, the adult legs and ventral body wall develop from larval tissues called leg imaginal discs. While the three pairs of legs—each present in the thoracic (T) segments T1, T2 and T3—have similar overall structures, they also have characteristic morphological differences unique to each leg pair[9,10] (Fig. 1a). These morphological differences are Hox-dependent, with Scr dictating T1 leg characteristics and Ubx dictating T3 leg characteristics[11]. Consistently, Scr is expressed in T1, but not T3, and Ubx is expressed in T3, but not T1, leg imaginal discs, the larval precursors of the adult appendages (Fig. 1b). Removing Scr function from a developing T1 leg or Ubx function from a developing T3 leg results in the homeotic transformation to a T2 leg fate (Fig. 1c)[11,12]. Consequently, all differences between T1 and T3 legs can be attributed either directly or indirectly to Scr and Ubx functions. Thus, the developing T1 and T3 legs provide a natural setting to compare Scr and Ubx functions in serially homologous tissues in vivo without the need for analyzing mutants.

To leverage this system under physiological conditions, we first compared the global transcriptomes of the T1 and T3 leg imaginal discs. Not surprisingly, these profiles are very similar to each other, with only a handful of genes, including Scr and Ubx, showing more than a two-fold difference in expression levels (Fig. 1d, Supplementary Fig. 1, and Supplementary Data 1). Next, genome editing was used to insert a 3×FLAG epitope tag at the endogenous Scr and Ubx loci (Fig. 1e), allowing us to use the same anti-FLAG antibody to obtain genome-wide binding data for both Hox paralogs (see "Methods", Supplementary Fig. 2 and ref. [13]). Multiple verified alleles for both genotypes (3×FLAG-Scr and 3×FLAG-Ubx) were homozygous viable and fertile, and did not show any noticeable developmental delays or defects.

### Genome-wide identification of paralog-specific and shared Hox-binding events

To determine if and to what extent paralog-specific Hox–DNA binding contributes to Scr- and Ubx-specific gene networks in the legs, ChIP-seq experiments against the 3×FLAG tag were performed from T1 and T3 leg imaginal discs dissected from 3×FLAG-Scr and 3×FLAG-Ubx lines, respectively, both isogenized into the same $w^{1118}$ genetic background. As a negative control, 3×FLAG ChIP-seq experiments were also performed using T1 leg discs from isogenic $w^{1118}$ flies with no FLAG epitope. Thousands of DNA-binding events were identified from both 3×FLAG-tagged Hox lines, whereas fewer than 20 were detected from T1 leg discs from the isogenic $w^{1118}$ line (Fig. 1f, Supplementary Fig. 4b, c). Examples of loci showing both similar and differential Scr binding in T1 and Ubx binding in T3 could be readily identified (Fig. 1f). For both 3×FLAG-Scr and 3×FLAG-Ubx, ~45% of loci were located in intergenic or intronic regions, consistent with binding to cis-regulatory modules (CRMs)[14,15] (Fig. 1g). Importantly, de novo motif searches identified Hox–Exd motifs that were remarkably consistent with those preferred by Scr-Exd and Ubx-Exd in vitro (Fig. 1h and Supplementary Fig. 3a)[8].

Genome-wide differential binding analysis was then performed to compare Scrbound loci in T1 leg discs (referred to as $Scr_{T1}$ loci) and Ubx bound loci in T3 leg discs (referred to as $Ubx_{T3}$ loci; see "Methods"). This analysis revealed that $Scr_{T1}$ and $Ubx_{T3}$ occupancy scores showed a strong positive correlation (Pearson's correlation coefficient: 0.706; Supplementary Fig. 4a). Consistently, a majority of binding events are shared between $Scr_{T1}$ and $Ubx_{T3}$ (referred to as $Scr_{T1} \approx Ubx_{T3}$ loci), while a subset of loci is

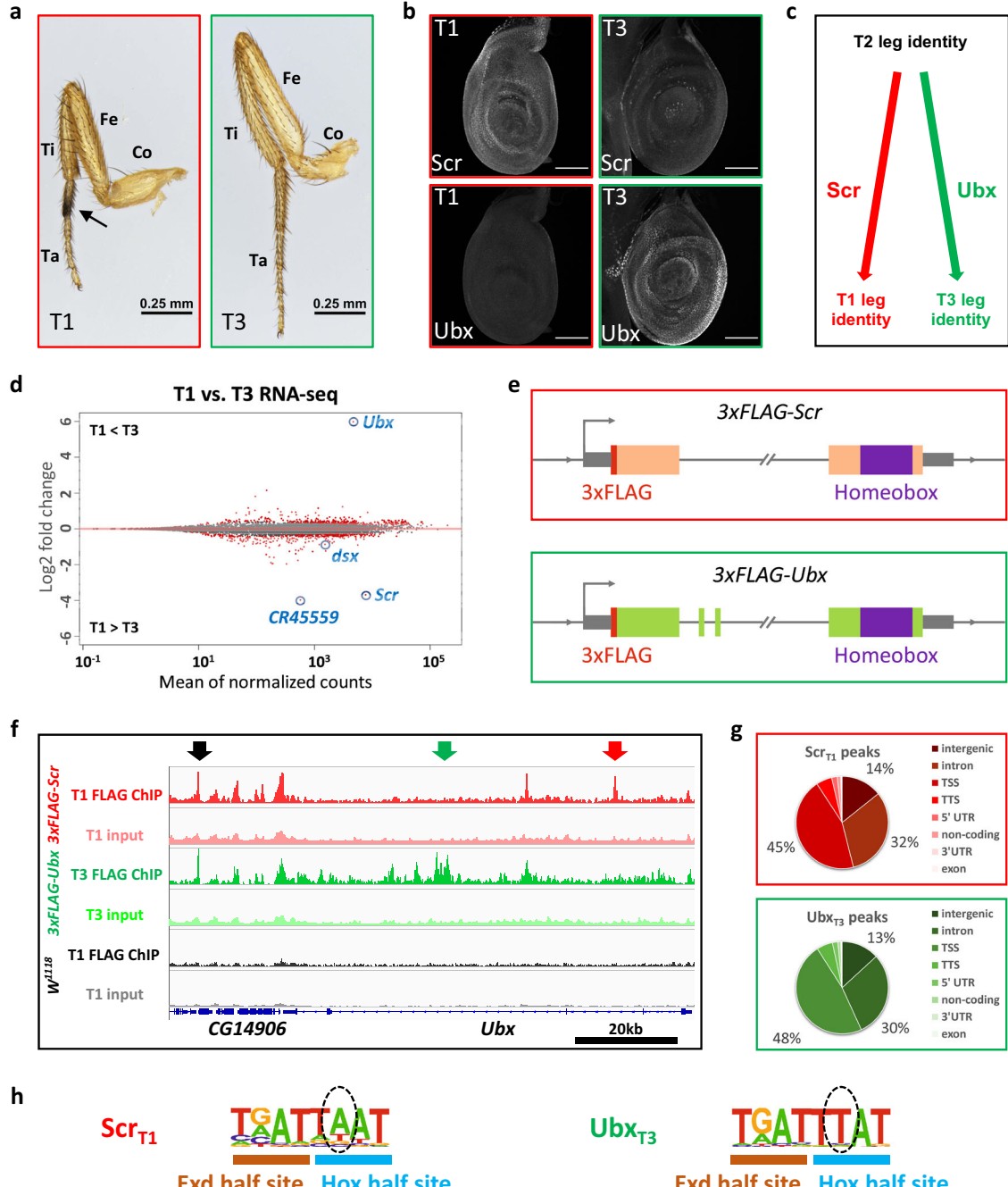

**Fig. 1 Genome-wide Scr and Ubx ChIP-seq and transcriptomes from T1 and T3 leg discs. a** Medial view of T1 and T3 legs from adult males. Co coxa, Fe femur, Ti tibia, Ta tarsus. Compared to T3, T1 legs have larger coxa and shorter femurs and, in males, have a tightly packed row of specialized bristles called sex combs on the first tarsal segment (arrow). **b** Co-immunostaining of Scr and Ubx proteins in T1 and T3 leg discs in the wandering larva stage. Weak Scr signal in T3 leg disc is from adepithelial cells[1]. Scale bar: 100 μm. **c** Summary of Scr and Ubx functions in specifying T1 and T3 leg identities, respectively. **d** MA plot comparing the T1 and T3 leg disc transcriptomes. Differentially expressed genes (FDR < 0.01) are labeled red. Several genes investigated in this study are indicated. *CR45559* is a lincRNA near the *Scr* locus. **e** Schematics (not to scale) of the 3xFLAG-tagged *Scr* and *Ubx* alleles generated by genome targeting ("Methods")[2]. The wide boxes (*Scr*, orange; *Ubx*, green) indicate coding regions; the homeobox is purple and N-terminal *3xFLAG* tags is red. The gray boxes are UTRs. The double-slash denotes large introns. The direction of transcription is indicated by an arrow at the transcription start site (TSS). **f** Genome browser view near the *Ubx* locus showing anti-FLAG ChIP-seq data from T1 or T3 leg discs dissected from isogenic stocks containing the *3xFLAG-Scr, 3xFLAG-Ubx*, or no *FLAG*-tagged allele (*w^1118*). Arrows indicate examples of different classes of binding: red: Scr$_{T1}$ > Ubx$_{T3}$, black: Scr$_{T1}$ ≈ Ubx$_{T3}$, green: Scr$_{T1}$ < Ubx$_{T3}$. **g** Pie graphs showing the genomic classification of Scr and Ubx ChIP-seq peaks. TSS: transcription start site (promoter), defined as -1 kb to +100 bp from the +1 nucleotide of mRNA. TTS transcription termination site. **h** Hox–Exd-binding motifs are the most significantly enriched motifs in Scr$_{T1}$ and Ubx$_{T3}$ peaks located in intergenic regions or introns (see Supplementary Fig. 3a for complete lists). Hox and Exd half-sites are indicated. Dashed ovals indicate positions that are known to differ for Scr-Exd and Ubx-Exd[3].

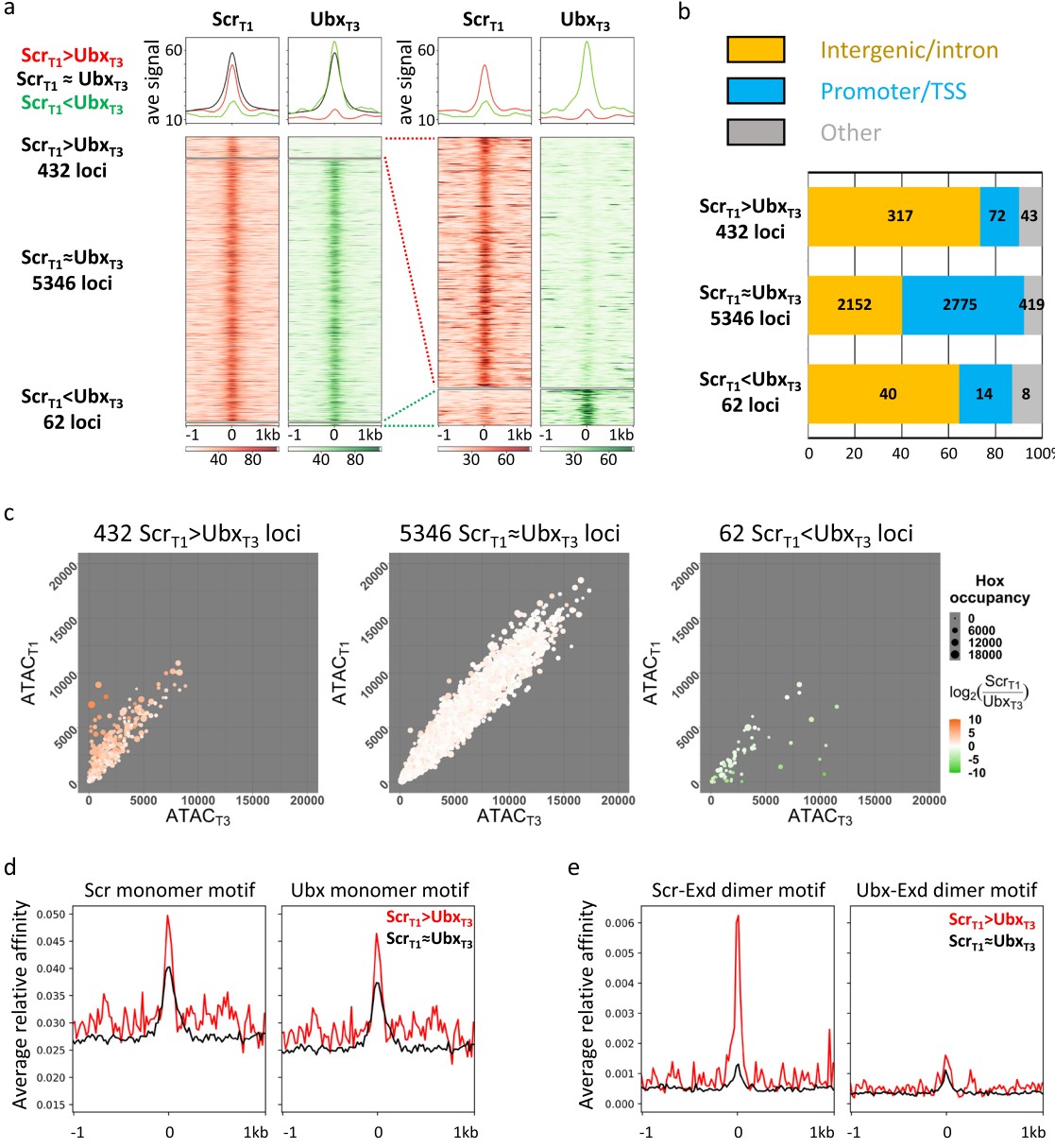

**Fig. 2 Genome-wide comparison between $Scr_{T1}$ and $Ubx_{T3}$ DNA-binding profiles. a** Left: heatmaps and histograms of $Scr_{T1} > Ubx_{T3}$, $Scr_{T1} \approx Ubx_{T3}$ and $Scr_{T1} < Ubx_{T3}$ loci plotted for $Scr_{T1}$ and $Ubx_{T3}$ ChIPs signals. Right: blow-up of the 432 $Scr_{T1} > Ubx_{T3}$ and 62 $Scr_{T1} < Ubx_{T3}$ loci. Loci in each of the three classes are sorted by the FDR values generated by DiffBind in ascending order (see "Methods" for details). The loci are aligned at the peak center, with $+/-1$ kb shown. In all heatmaps in this and other panels, the color intensity scores are arbitrary values indicating relative TF occupancy at the target locus. **b** Bar graph showing the genome region classification of $Scr_{T1} > Ubx_{T3}$, $Scr_{T1} \approx Ubx_{T3}$ and $Scr_{T1} < Ubx_{T3}$ loci. **c** Scatter plots comparing T1 and T3 chromatin accessibility in $Scr_{T1} > Ubx_{T3}$ (left), $Scr_{T1} \approx Ubx_{T3}$ (middle) and $Scr_{T1} < Ubx_{T3}$ (right) loci. The size of a dot represents the average of $Scr_{T1}$ and $Ubx_{T3}$ ChIP signals at that locus, and the color indicates the log2 ratio between $Scr_{T1}$ and $Ubx_{T3}$ ChIP-seq signals. **d, e** Histograms showing relative affinity scores using *NRLB* models for Scr and Ubx monomers (**d**), and Scr-Exd and Ubx-Exd heterodimers (**e**) in $Scr_{T1} > Ubx_{T3}$ (red) and $Scr_{T1} \approx Ubx_{T3}$ (black) loci $+/-1$ kb relative to the peak center.

strongly biased towards either Scr or Ubx (referred to as $Scr_{T1} > Ubx_{T3}$ and $Scr_{T1} < Ubx_{T3}$ loci, respectively; Fig. 2a, see "Methods"). Notably, there are approximately sevenfold more $Scr_{T1} > Ubx_{T3}$ loci than $Scr_{T1} < Ubx_{T3}$ loci, suggesting a strong asymmetry in the number of paralog-specific targets. Compared to $Scr_{T1} \approx Ubx_{T3}$ loci, paralog-specific ones are more likely to be intergenic or intronic, suggesting that CRMs are enriched in paralog-specific loci (Fig. 2b).

**Most paralog-specific loci do not show differences in chromatin accessibility.** Our results so far reveal that Scr and Ubx

have both paralog-specific and shared targets in T1 and T3 leg discs. Before addressing the functions of these paralog-specific binding events (see below), we first asked how Scr and Ubx bind to their paralog-specific targets in vivo, despite having very similar DNA-binding properties in vitro[5–7]. In some cases, differential chromatin accessibility underlies tissue-specific gene regulation[16], we therefore determined the degree to which chromatin accessibility can account for paralog-specific Hox binding in the leg imaginal discs, using ATAC-seq[17]. We identified ~20,000 accessible loci in T1 and T3 leg discs (referred to as $ATAC_{T1}$ and $ATAC_{T3}$). Generally, the chromatin accessibility profiles of the two leg discs are highly similar (Supplementary

Fig. 4d), with little correlation between either $Scr_{T1}$ binding and T1 accessibility or $Ubx_{T3}$ binding and T3 accessibility (Supplementary Fig. 4b, c). The handful of loci that are more accessible in one disc are biased towards binding the Hox protein expressed in that disc (Fig. 2c), and those exhibiting the most significant difference in accessibility are located in either the *Antennapedia* complex, where *Scr* resides, or the *bithorax* complex, where *Ubx* is located (Supplementary Data 2). We also examined the chromatin accessibility in the T2 leg disc and found that it is also very similar to the T1 and T3 profiles (Supplementary Fig. 4d). The similar ATAC-seq profiles for all three pairs of leg discs suggest that chromatin accessibility is neither altered by Hox expression nor can it account for paralog-specific Hox–DNA binding.

**Relative affinities of Hox–Exd dimers, but not Hox monomers, correlate with ChIP-seq patterns.** To determine the extent to which paralog-specific binding can be explained by the intrinsic DNA-binding specificities of Hox monomers or Hox–Exd dimers, we used *No Read Left Behind* (*NRLB*)[18], a computational method that transforms in vitro SELEX-seq data into models capable of capturing a TF's binding specificity over its entire affinity range. The $Scr_{T1} \approx Ubx_{T3}$ loci have a similar normalized mean relative affinity enrichment score for Scr and Ubx monomer binding near the peak center (Fig. 2d). A similar pattern of relative affinity enrichment for both monomers is also observed for $Scr_{T1} > Ubx_{T3}$ loci, suggesting that the intrinsic DNA-binding specificities of Scr and Ubx monomers cannot account for $Scr_{T1} > Ubx_{T3}$ binding. In contrast, there is strong differential enrichment only for the Scr-Exd motif score in $Scr_{T1} > Ubx_{T3}$ loci (Fig. 2e). $Scr_{T1} < Ubx_{T3}$ loci were not analyzed due to their small number. These results suggest that the latent specificity conferred by heterodimerization between Scr and Exd significantly contributes to paralog-specific Hox binding genome-wide.

**Generation of an Scr mutant that is unable to interact with Exd.** To definitively test if Exd heterodimerization contributes to paralog-specific binding, we generated an *Scr* allele that expresses a mutant protein unable to interact with Exd. We chose to mutate Scr for two reasons: first, there were more $Scr_{T1} > Ubx_{T3}$ loci than $Scr_{T1} < Ubx_{T3}$ loci (Fig. 2a); and second, there is only a single Exd-interacting W-motif in Scr (Fig. 3a), whereas multiple Exd-interaction motifs are present in Ubx[19,20]. The resulting *3×FLAG-Scr(YPWM\*)* allele expresses a 3×FLAG-tagged Scr protein with its YPWM motif mutated to four alanines (Fig. 3b and Supplementary Fig. 2, see "Methods"). As with our other edited alleles, multiple independent isolates of this mutant were isogenized into the *w^1118* genetic background.

The *3×FLAG-Scr(YPWM\*)* allele is lethal as a homozygote, demonstrating that Scr's YPWM motif is essential for viability. However, this allele only impairs a subset of known *Scr* functions. For example, homozygous *Scr(YPWM\*)* embryos fail to express *CrebA*, a known Scr target[21] in salivary glands (Fig. 3c). In contrast, although the number of sex combs on the male T1 leg is reduced in *Scr* null/+ heterozygous animals, the number of sex combs is unaffected in heterozygotes of the *Scr(YPWM\*)* allele (Fig. 3d). The lack of an effect on sex comb number makes sense because these structures are derived from a part of the leg disc that does not express Homothorax (Hth), a transcription factor required for Exd's nuclear localization[22] and thus its function as a Hox cofactor (see below). A third well-characterized *Scr* function is the suppression of the sternopleural (Sp) bristles, which are normally present in T2 but not in T1 legs[23]. Suppression of these bristles remains intact in T1 legs containing homozygous clones of the *Scr(YPWM\*)* allele (Fig. 3e). In contrast to the sex combs, the precursors of the Sp bristles[23], revealed by the expression of

Achaete (Ac), are derived from a region of the T2 leg disc where Hth is expressed and Exd is nuclear (Fig. 3f). Thus, even in cells where Exd is nuclear, Scr can execute paralog-specific functions in an Exd-independent manner.

**Scr-Exd interaction is required for a subset of Scr–DNA-binding events in vivo.** To assess the importance of the Scr-Exd interaction genome-wide, we performed ChIP-seq experiments with the 3×FLAG-Scr(YPWM\*) protein. Due to the homozygous lethality of the *3×FLAG-Scr(YPWM\*)* allele, these ChIPs were done with T1 leg discs from *3×FLAG-Scr(YPWM\*)/+* heterozygous animals (see "Methods"). For comparison, anti-FLAG ChIP-seq experiments were also carried out using T1 leg discs from *3×FLAG-Scr/+* heterozygous larvae. This analysis identified three sets of Scr-bound loci (Fig. 4a, d). The largest set is comprised of thousands of loci where the WT and mutant Scr proteins bind similarly ($Scr_{T1} \approx Scr(YPWM^*)_{T1}$). We infer these loci to be Exd-independent, because binding does not require the YPWM motif. The second set is comprised of hundreds of loci where the mutant protein binds significantly more poorly than WT Scr ($Scr_{T1} > Scr(YPWM^*)_{T1}$). We infer these loci to be Exd-dependent, as they require Scr's YPWM motif, and Exd is the only known protein to interact with this motif. The third and smallest set is comprised of loci where mutant occupancy is greater than WT occupancy (Fig. 4d). This third set may reflect the YPWM mutant's enhanced ability to bind monomeric Hox-binding sites (which are AT-rich) within accessible regions of the genome. Consistent with this notion, binding of the YPWM mutant is skewed towards AT-rich promoter/TSS regions (Fig. 4b), and the $Scr_{T1} < Scr(YPWM^*)_{T1}$ loci score strongly for the Scr monomer *NRLB* model (Fig. 4d).

Because the nuclear localization and DNA binding of Exd depend on its interaction with Hth[22], Hth ChIP-seq data can be used to infer the genome-wide binding of Exd. Notably, Hth ChIP-seq from T1 leg discs reveal higher Hth occupancy in $Scr_{T1} > Scr(YPWM^*)_{T1}$ loci compared to $Scr_{T1} \approx Scr(YPWM^*)_{T1}$ loci, providing independent evidence that the $Scr_{T1} > Scr(YPWM^*)_{T1}$ loci are also bound by Exd. Further, de novo motif discovery identified Hox–Exd-binding motifs in $Scr_{T1}$, but not $Scr(YPWM^*)_{T1}$ bound peaks (Fig. 4c and Supplementary Fig. 3b). Finally, using our *NRLB* models, the Scr-Exd dimer motif scores are significantly stronger in $Scr_{T1} > Scr(YPWM^*)_{T1}$ loci, compared to $Scr_{T1} \approx Scr(YPWM^*)_{T1}$ loci. In contrast, scoring for the Scr monomer motif reveals a weak signal near the peak centers of both $Scr_{T1} > Scr(YPWM^*)_{T1}$ loci and $Scr_{T1} \approx Scr(YPWM^*)_{T1}$ loci (Fig. 4d).

**Scr-Exd interaction is required for many, but not all $Scr_{T1} > Ubx_{T3}$ binding events.** The above results demonstrate that the binding of Scr to many of its in vivo targets requires its YPWM motif, consistent with a requirement for heterodimerization with Exd. We next asked to what extent do these Exd-dependent loci account for the $Scr_{T1} > Ubx_{T3}$ loci described above (Fig. 2a). As a first step, all 432 $Scr_{T1} > Ubx_{T3}$ loci were ordered by their Exd-dependency, based on the $Scr_{T1}$ and $Scr(YPWM^*)_{T1}$ ChIP-seq results (see "Methods" for details). Accordingly, the sites that depend the most on Exd are at the top of these heatmaps, while the sites at the bottom are the most Exd-independent. Of the 432 $Scr_{T1} > Ubx_{T3}$ loci, 141 are high confidence Exd-dependent Scr-specific loci, while 172 are high confidence Exd-independent loci (see "Methods"). Compared to their percentage in all Scr-bound loci (Fig. 4d), Exd-dependent loci constitute a higher portion among $Scr_{T1} > Ubx_{T3}$ loci, consistent with the notion that heterodimerization with Exd significantly contributes to paralog-specific Scr–DNA binding in vivo.

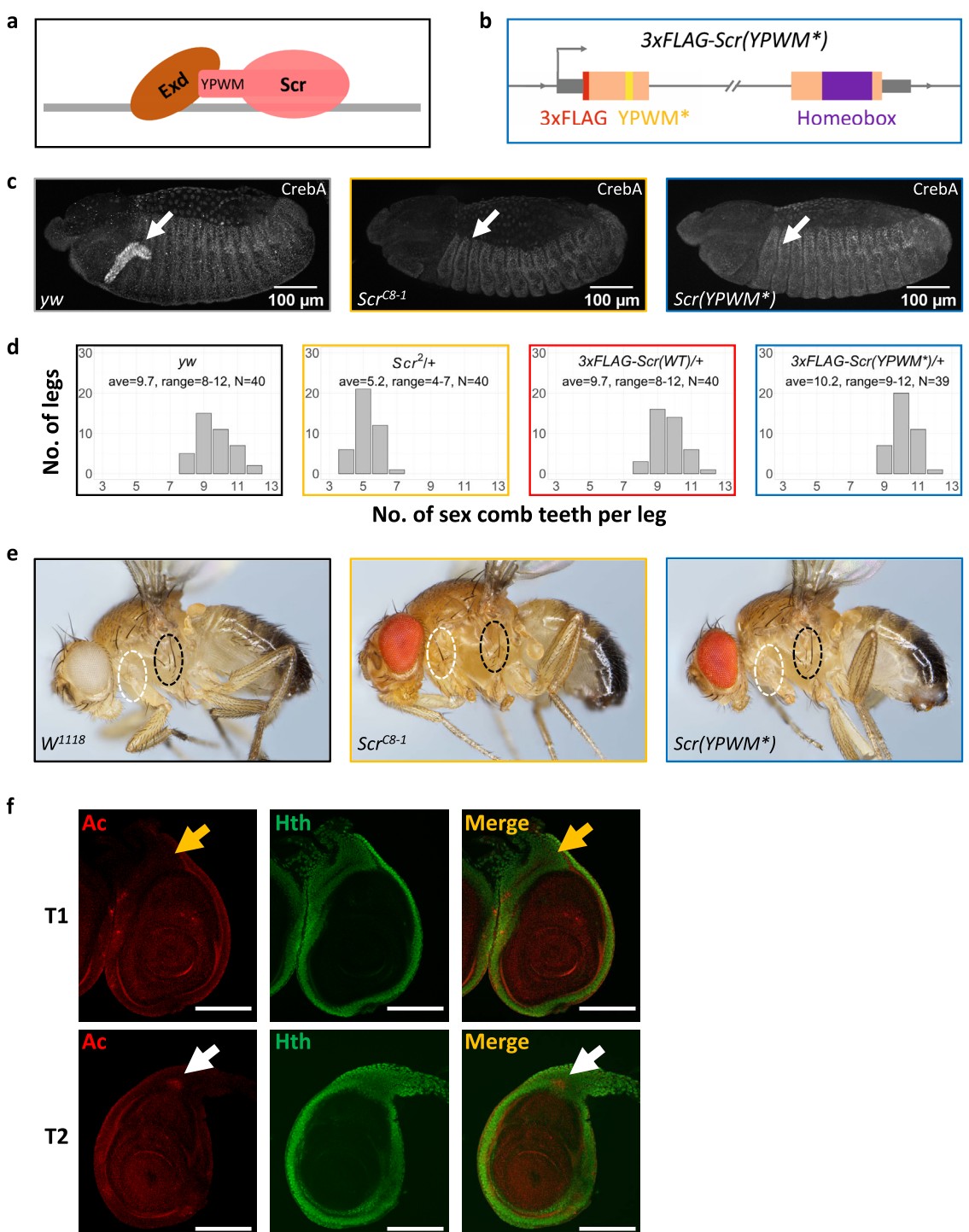

Moreover, Hth occupancy at $Scr_{T1} > Ubx_{T3}$ loci shows a positive correlation with their Exd-dependency (Fig. 5a). Scr monomer and Scr-Exd dimer relative affinities, predicted by *NRLB*, also display the expected correlation with Exd-dependency: there is a clear Scr monomer signature in Exd-independent peaks, while a strong Scr-Exd heterodimer signature is observed only among the Exd-dependent loci (Fig. 5a).

In summary, although a large fraction of the $Scr_{T1} > Ubx_{T3}$ loci requires Scr's YPWM motif, suggesting they are Exd-dependent, these results also indicate that there are Exd-independent mechanisms for Hox proteins to achieve paralog-specific DNA binding in vivo. They also suggest that

dependency on Exd is not an all-or-nothing phenomenon because, depending on the locus, Scr binding requires its YPWM motif to different degrees.

**Testing the function of paralog-specific Hox binding**. Our analyses so far reveal that Exd plays an important role in the binding of Scr to its paralog-specific targets, i.e., the $Scr_{T1} > Ubx_{T3}$ loci. Ultimately, it is differential gene expression that differentiates T1 and T3 leg identities. Therefore, we next investigated to what extent paralog-specific Scr–DNA binding translates into T1 ≠ T3 target gene transcription.

**Fig. 3 Generation and phenotypic characterization of a YPWM-mutated Scr allele. a** Schematic showing Scr-Exd interaction mediated by Scr's YPWM motif. The gray bar denotes DNA. **b** Schematic (not to scale) of the *3xFLAG-Scr(YPWM\*)* allele. Colors are defined in Fig. 1; the YPWM- > AAAA mutation (YPWM\*) is highlighted in yellow. The direction of transcription is indicated by an arrow at the TSS. **c** Left: CrebA is expressed in the embryonic salivary gland in wild-type embryos. Middle: CrebA expression in salivary gland is absent in homozygous *Scr* null embryos. Right: CrebA is also absent in the salivary gland of homozygous *Scr(YPWM\*)* embryos. The arrows point to the position of wild-type CrebA expression domain. **d** The number of sex comb teeth in males of various genotypes. The *x* axis is the number of sex comb teeth per leg, and the *y* axis shows the number of legs. The average and the range of sex comb teeth numbers for each genotype are also shown. **e** The suppression of sternopleural bristles (Sp bristles) in the T1 segment does not require Scr's YPWM motif. Left: wild type. Sp bristles are in T2 but not T1 segment. Middle: adult with homozygous *Scr* null clones in T1. Consistent with previous findings[4], Sp bristles are observed when *Scr* is absent (22 out of 28 adults with the correct genotype have Sp bristles in T1; see "Methods" for details). Right: adult with homozygous *Scr(YPWM\*)* clones in T1 segment. The Sp bristles are not observed (0 out of 16 adults with the correct genotype have Sp bristles in T1). White and black ovals indicate the equivalent regions of the T1 and Sp-bearing T2 segments, respectively. **f** Co-immunostaining of Ac, a marker for proneural clusters, and Hth proteins in T1 and T2 leg discs in the wandering larva stage. White arrows point to the Ac+ proneuronal clusters for the Sp bristles in the Hth domain of T2 leg discs, and yellow arrows point to the Ac- homologous positions in T1 leg discs, where the proneural cluster is suppressed by Scr. Scale bar: 100 μm.

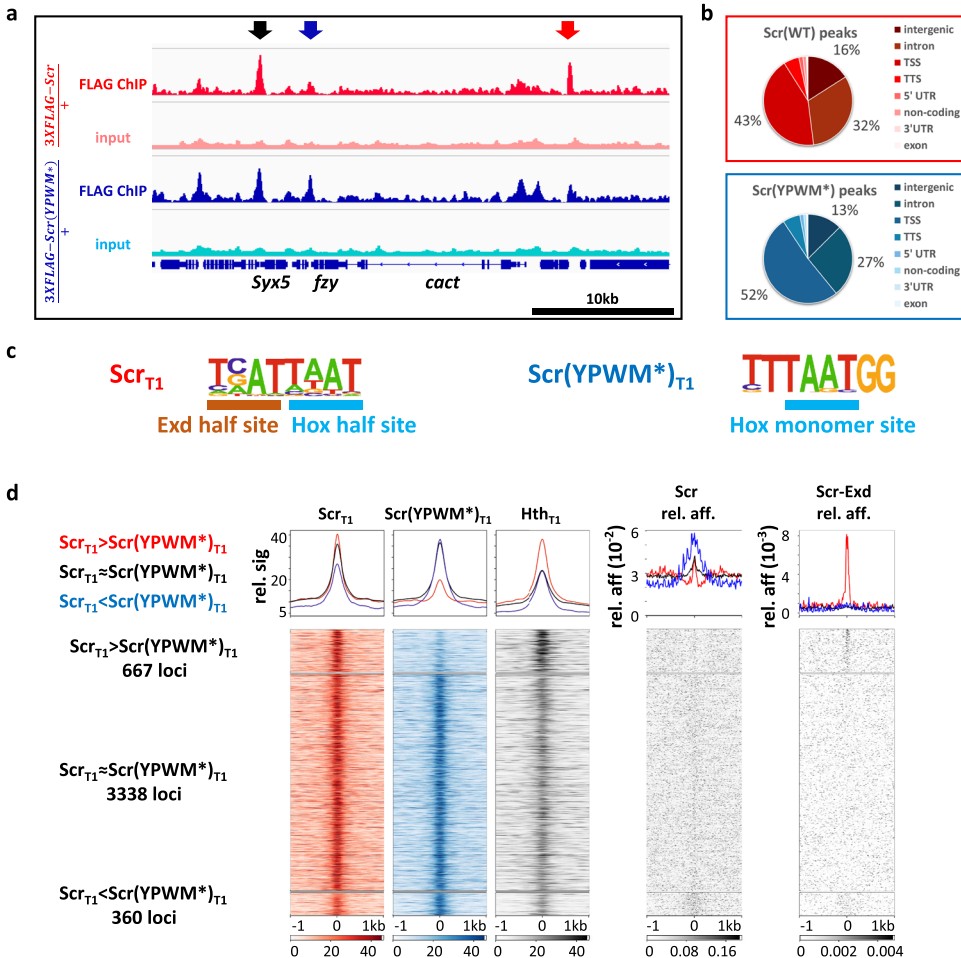

**Fig. 4 Genome-wide comparison of Scr$_{T1}$ and Scr(YPWM\*)$_{T1}$-binding profiles. a** Genome browser view near the *cact* locus showing anti-FLAG ChIP-seq data from T1 leg discs dissected from isogenic stocks heterozygous for the *3xFLAG-Scr* or *3xFLAG-Scr(YPWM\*)* alleles. Arrows indicate examples of different classes of peaks: red: Scr$_{T1}$ > Scr(YPWM\*)$_{T1}$, black: Scr$_{T1}$ ≈ Scr(YPWM\*)$_{T1}$, blue: Scr$_{T1}$ < Scr(YPWM\*)$_{T1}$. **b** Pie graphs showing the genome region classification of Scr$_{T1}$ and Scr(YPWM\*)$_{T1}$ ChIP-seq peaks. **c** Scr$_{T1}$ and Scr(YPWM\*)$_{T1}$ peaks located in intergenic and intronic regions are enriched for Exd-Scr heterodimer and Scr monomer binding sites, respectively. Hox and Exd half-sites are indicated in the heterodimer motif. See Supplementary Fig. 3a for complete lists. **d** Heatmaps and histograms of Scr$_{T1}$ > Scr(YPWM\*)$_{T1}$, Scr$_{T1}$ ≈ Scr(YPWM\*)$_{T1}$ and Scr$_{T1}$ < Scr(YPWM\*)$_{T1}$ loci plotted for Scr$_{T1}$, Scr(YPWM\*)$_{T1}$, and Hth$_{T1}$ ChIP-seq signals. The relative affinities from *NRLB* models of Scr monomer and Scr-Exd heterodimer are shown to the right. Loci in each of the three classes are sorted by the FDR values generated by DiffBind in ascending order. Loci are aligned at the peak center, with +/−1 kb shown.

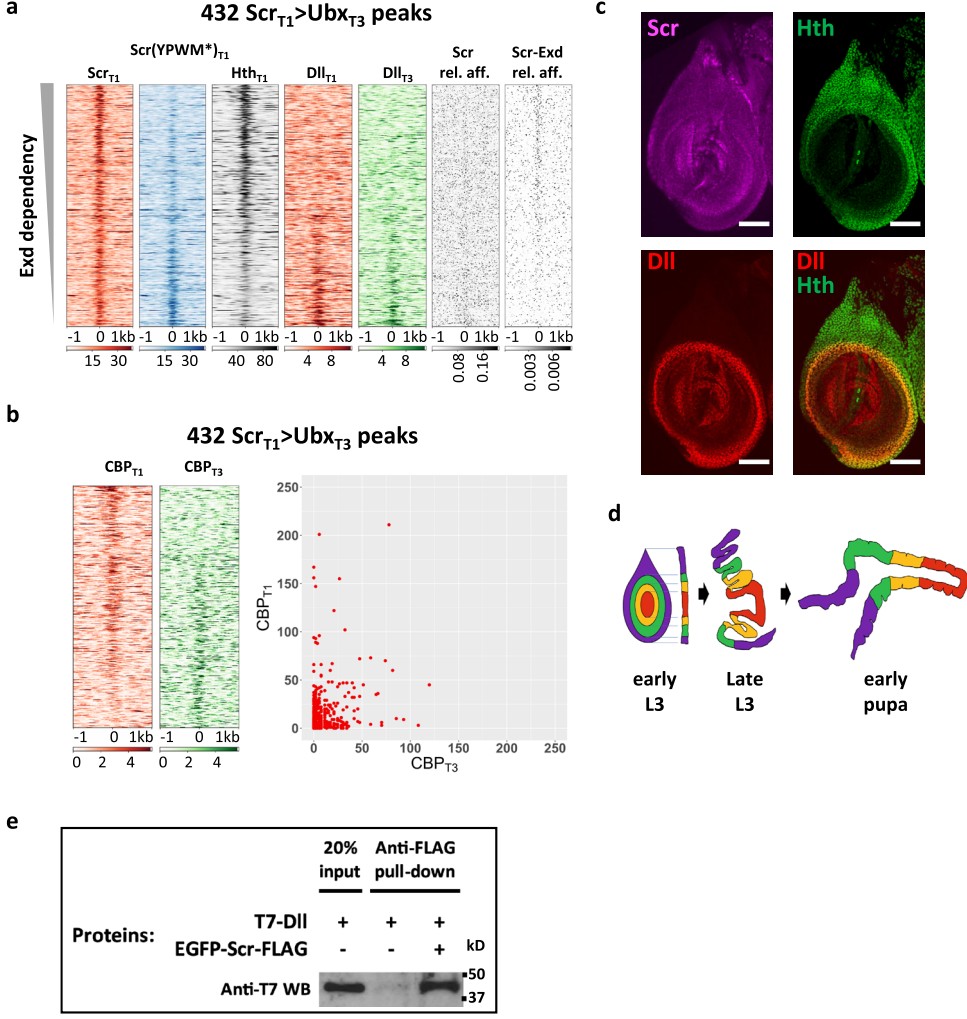

**Fig. 5 Exd-dependent and -independent Scr$_{T1}$ > Ubx$_{T3}$ peaks. a** Heatmaps of the 432 Scr$_{T1}$ > Ubx$_{T3}$ ChIP-seq peaks sorted by Exd-dependency, based on the relative intensities of the Scr$_{T1}$ and Scr(YPWM*)$_{T1}$ ChIP-seq signals. Also shown are the ChIP-seq signal for Hth in T1 leg disc and the ChIP-seq signals for Dll in T1 and T3 leg discs. The relative affinities of Scr monomer (Scr) and Scr-Exd dimers using the respective *NRLB* models are also plotted.
**b** Heatmaps (left) and scatter plot (right) showing CBP occupancy at the 432 Scr$_{T1}$ > Ubx$_{T3}$ peaks in T1 and T3 leg discs. The peaks in the heatmaps are sorted according to the ratio of T1:T3 CBP occupancy. **c** The expression patterns of Scr (magenta), Hth (green), and Dll (red) in T1 leg imaginal discs. Scale bar: 50 μm. **d** Schematic showing the morphological changes of leg discs during metamorphosis. Both top view and lateral cross-section views are shown for the early L3 stage, and the lateral cross-section view is shown for late L3 (wandering stage) and early pupal stages. **e** Co-immunoprecipitation showing the physical interaction between Scr and Dll. Immunoprecipitation was performed using anti-FLAG antibody and western blot was probed with anti-T7 antibody. This experiment was repeated three times, and one representative result is shown. Source data are provided as a Source Data file.

We first performed ChIP-seq against Creb binding protein (CBP), a known marker for CRM activity[24], in T1 and T3 leg discs. In Scr$_{T1}$ > Ubx$_{T3}$ loci, CBP occupancy ranges from significantly T1 > T3 to markedly T1 < T3, and those loci with the highest CBP signals tend to have T1 > T3 CBP occupancy (Fig. 5b). The presence of CBP at Scr$_{T1}$ > Ubx$_{T3}$ loci suggests that they are indeed active CRMs. In addition, the observation that among the Scr$_{T1}$ > Ubx$_{T3}$ loci CBP can be biased to either T1 or T3 leg discs suggests that Scr may act both to repress transcription (when CBP$_{T1}$ < CBP$_{T3}$) and to activate transcription (when CBP$_{T1}$ > CBP$_{T3}$). Such a context-dependent pattern of CRM activity is consistent with the expectation for a selector transcription factor-like Scr.

We next examined the expression of genes near Scr-bound loci (Supplementary Fig. 5). We found that genes near Scr$_{T1}$ > Ubx$_{T3}$ loci are more likely to be expressed in a T1 > T3 pattern, which agrees with the CBP occupancy pattern above. Interestingly, genes near Scr$_{T1}$ > Scr(YPWM*)$_{T1}$ loci also tend to show a T1 > T3

expression pattern (Supplementary Fig. 5), suggesting that the Exd-dependent Scr target CRMs tend to be enhancers, as opposed to silencers. This is consistent with previous work suggesting that Hth functions as a transcription activator in vivo[25].

Finally, to estimate how many of the Scr$_{T1}$ > Ubx$_{T3}$ events leads to T1 ≠ T3 CRM activity, we generated *lacZ* reporter genes from twenty-five Exd-dependent Scr$_{T1}$ > Ubx$_{T3}$ loci (see "Methods"). Eight out of the twenty-five selected loci (~1/3) drove T1 > T3 expression, while the rest either drove T1 = T3 expression (11 of 25), or were not active in 3rd instar leg discs (6 of 25).

In general, one or two putative Hox–Exd-binding motifs were readily identified near the Scr$_{T1}$ ChIP peak center of each selected CRM, consistent with the pattern of Scr-Exd dimer motifs shown in Fig. 5a. When these Hox–Exd motifs were mutated (4 bp substitutions at the center of the motifs, see "Methods" for details) in three selected T1 > T3 CRMs (*ac-1*, *h-1*, and *fj-1*), all three lost expression in leg discs (Fig. 6a–c), indicating direct regulation by Scr-Exd. In contrast, when the highest affinity

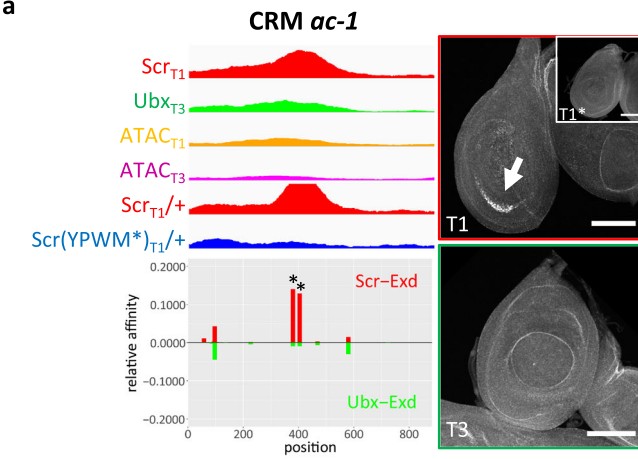

## CRM ac-1

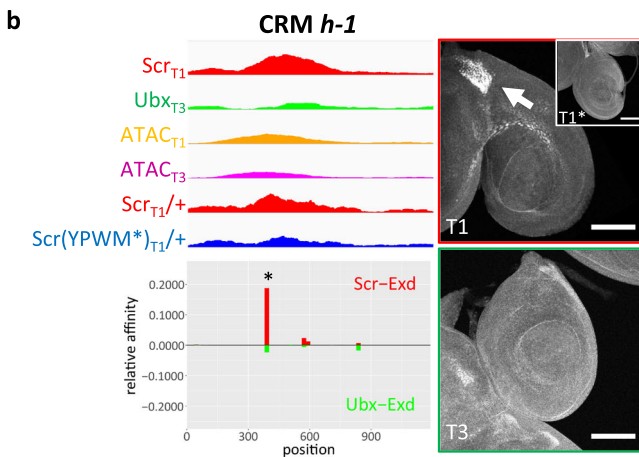

## CRM h-1

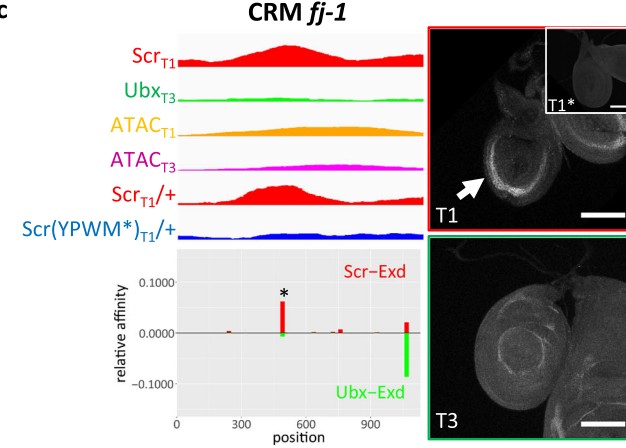

## CRM fj-1

**Fig. 6 Reporters generated from Exd-dependent $Scr_{T1} > Ubx_{T3}$ peaks.**
Examples of Exd-dependent $Scr_{T1} > Ubx_{T3}$ peaks that drive T1 > T3 reporter expression patterns in leg discs, from loci near *ac* (*ac-1*; **a**), *h* (*h-1*; **b**), and *fj* (*fj-1*; **c**). Each CRM characterized in this study was named after a nearby gene. For each fragment covering the selected peak, genome browser tracks for $Scr_{T1}$, $Ubx_{T3}$, $Scr_{T1}/+$ and $Scr(YPWM^*)_{T1}/+$ ChIP-seq signals, as well as $ATAC_{T1}$ and $ATAC_{T3}$ signals, are shown and below them are the *NRLB* relative affinity predictions for Scr-Exd (red bars) and Ubx-Exd (green bars). The relative affinity tracks are aligned to the ChIP and ATAC tracks. The binding sites chosen for mutagenesis are close to the center of the Scr ChIP-seq peak and are indicated with asterisks. Immunostains showing reporter expression in T1 and T3 leg discs for the wild-type reporters are shown to the right; the insets show the expression in T1 discs of reporters where the Scr-Exd-binding sites were mutated. Scale bar: 100 μm.

**Dll is a candidate Scr cofactor in the distal leg domain.** As mentioned above, Exd is not present in the nuclei of all leg disc cells, due to the restricted expression of Hth in the periphery of the leg discs (Fig. 5c) that gives rise to the ventral body wall and proximal segments of the adult legs (Fig. 5d). However, there are well-documented Hox-dependent morphological differences between the T1 and T3 legs in the distal domain[26], suggesting that Hox proteins must execute a subset of paralog-specific functions in an Exd-independent manner. Since our results suggest that Hox monomer binding is unlikely to account for paralog-specific in vivo binding (Fig. 2d), we hypothesized that there must be additional distal cofactor(s) that contribute to paralog-specific binding and activity.

Notably, the transcription factors Teashirt (Tsh)[27] and Distal-less (Dll)[28] have been shown to physically interact with Scr, and there is evidence that Disconnected (Disco)[29] genetically interacts with Scr. In addition, Engrailed (En) and Sloppy-paired (Slp)[30] have been shown to interact with the abdominal Hox proteins Ubx and Abd-A. Among these candidates, the homeodomain containing transcription factor Dll stood out because it is expressed in the distal domain of the leg discs, which is largely complementary to the Hth-expressing domain (Fig. 5c), and is known to be important for the identity of the distal leg[31,32], in part by repressing *hth* expression[33]. Consistent with previous bimolecular complementation (BiFC) results[28], we also confirm that Scr and Dll physically interact by co-immunoprecipitation (Fig. 5e).

To initially examine a role for Dll in Hox–DNA binding, we carried out ChIP-seq experiments for Dll in T1 and T3 leg discs. We observed a striking correlation in which Exd-independent, but not Exd-dependent $Scr_{T1} > Ubx_{T3}$ loci have a strong tendency to bind Dll (Fig. 5a). A similar Dll occupancy gradient is also seen in T3 leg discs (Fig. 5a). These observations are consistent with a model in which Dll is a Hox cofactor in the distal leg, in cells where Exd is not available as a cofactor.

**Dll facilitates Scr–DNA binding in a sequence-specific manner.** If Dll is a bona fide Hox cofactor in the T1 leg disc, we would expect that Scr and Dll may promote each other's binding to a subset of DNA sequences. To identify sequences bound by Scr +Dll in an unbiased manner, we used a gel-free SELEX protocol (see "Methods"). SELEX libraries were generated and sequenced for Scr, Dll, Scr-Exd (as a positive control), and Scr-Dll, and *NRLB*-binding models were generated (Fig. 7a–d).

The Scr and Scr-Exd models agree well with models generated from previous SELEX data, which used electrophoretic mobility shift assays (EMSAs) to isolate protein-bound DNAs (Fig. 7a, c)[18]. The model obtained with Dll is consistent with binding by a homeodomain (Fig. 7b). The Scr-Dll *NRLB* model shows a

Hox–Exd motifs were mutated in three selected CRMs that did not drive T1 > T3 expression (2 of them have T1 = T3 expression and one has no expression in leg discs), none showed any detectable change in reporter expression. Thus, although Scr is bound to these CRMs, these motifs are not required for CRM activity.

The reporter gene results suggest that when the tissue-specific activity of a CRM agrees with the paralog-specific Hox ChIP pattern at that CRM, it is likely to be directly regulated by Hox proteins. Extrapolating from these examples, we estimate that ~1/3 of all paralog-specific Hox-binding events at CRMs directly regulate their activity, and lead to paralog-specific regulation of transcription.

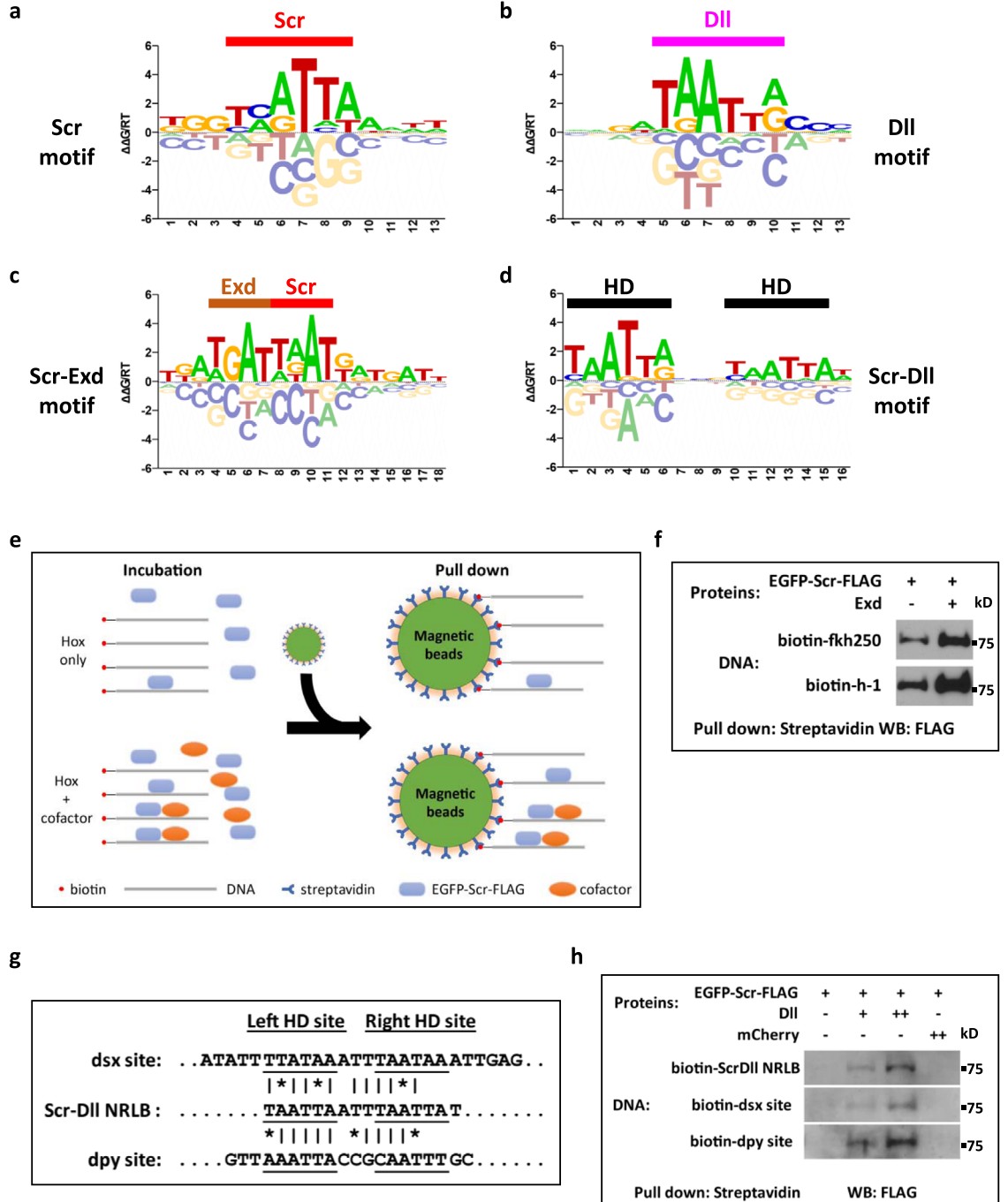

**Fig. 7 Characterization of Scr-Dll DNA binding preferences. a–d** *NRLB* models generated from gel-free SELEX datasets. **a** Scr monomer model. **b** Dll monomer model. **c** Scr-Exd dimer model. **d** Scr-Dll dimer model. Half-sites are indicated in Scr-Exd (**c**) and Scr-Dll (**d**) dimer models. **e** Schematic showing the in vitro gel-free pull-down assay to assess multi-TF-DNA binding. **f** Assay validation by testing the binding of Scr to the *fkh250* and *h-1* probes in the absence and the presence of Exd. This experiment was repeated three times and one representative result is shown. **g** Sequence alignment of the Scr-Dll *NRLB* consensus motif, the *dsx-1*, and the *dpy-1* probes. **h** Binding of Scr to DNA sequences containing the Scr-Dll *NRLB* consensus motif and the genomic fragments containing the *dsx-1* and *dpy-1* peaks. Binding was assessed in the absence and presence of Dll, and in the presence of a negative control protein mCherry. The *dsx-1* and *dpy-1* probes are derived from the center of the relevant Hox ChIP-seq peak (see Fig. 8). All experiments were repeated at least 3 times and one representative result is shown. Source data are provided as a Source Data file.

pattern of two homeodomain monomer binding motifs separated by a spacer of a few base pairs (Fig. 7d). This configuration is distinct from that of Hox–Exd heterodimers (Fig. 7c, d), in which the Hox and Exd half-sites partially overlap to form a composite binding motif.

EMSAs were unsuccessful at visualizing a DNA-bound Scr-Dll heterodimer, perhaps due to the non-physiological TBE buffer used in these assays. Instead, a gel-free pull-down assay that uses more physiologically relevant buffer conditions was performed to characterize Scr-Dll-DNA binding (Fig. 7e and "Methods"). Briefly, a biotin-labeled DNA probe is incubated with FLAG-tagged Scr protein, with and without either Exd or Dll. After pull-down with magnetic streptavidin beads, the DNA-bound Scr protein is visualized by western blot.

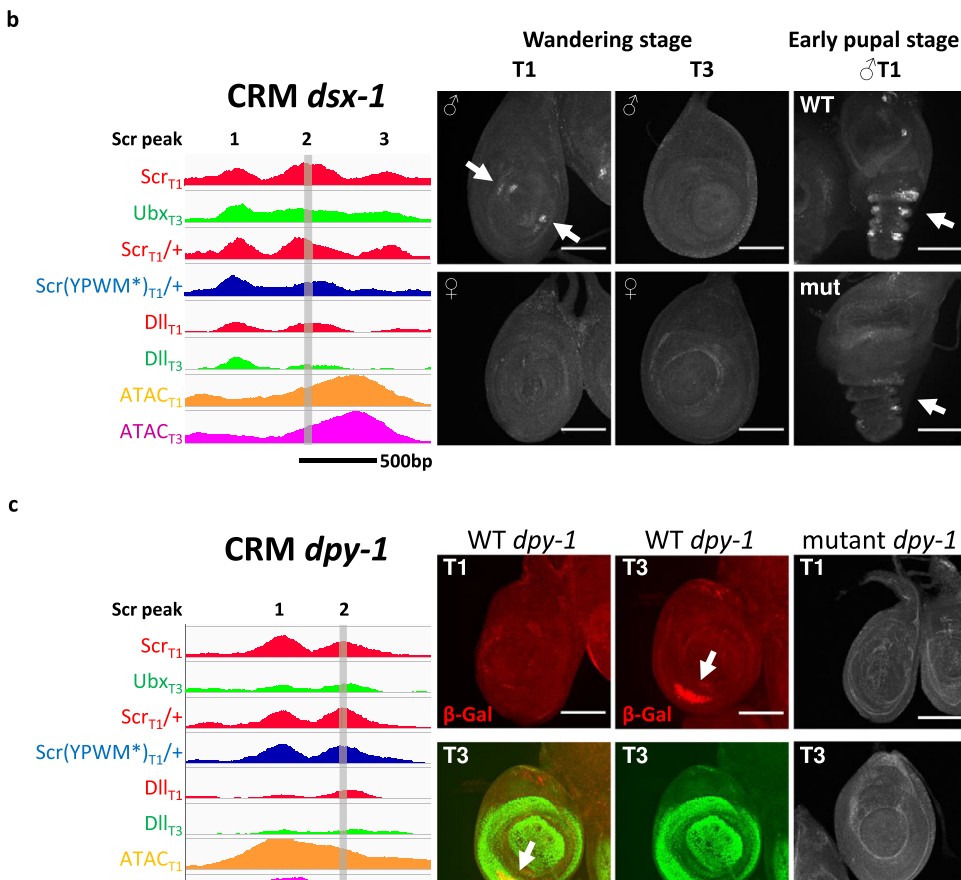

| Exd-independent *lacZ* reporter summary | | Expression pattern | | | | |
|---|---|---|---|---|---|---|
| | | T1>T3 | T1<T3 | T1=T3 | No expression | total |
| Scr$_{T1}$>Ubx$_{T3}$ and Scr$_{T1}$ ≈ Scr(YPWM*)$_{T1}$ | With Dll co-occupancy | 1 | 1 | 7 | 0 | 9 |
| | Without Dll co-occupancy | 3 | 0 | 2 | 1 | 6 |

**Fig. 8 Reporters generated from Exd-independent Scr$_{T1}$ > Ubx$_{T3}$ peaks. a** Table summarizing the leg disc expression patterns driven by selected Exd-independent Scr$_{T1}$ > Ubx$_{T3}$ CRMs. **b**, **c** Examples of Dll bound Exd-independent Scr$_{T1}$ > Ubx$_{T3}$ peaks from *dsx* (*dsx-1*; **b**) and *dpy* (*dpy-1*; **c**) that drive T1 ≠ T3 expression patterns in leg discs. On the left are genome browser tracks for the Scr$_{T1}$, Ubx$_{T3}$, Scr$_{T1}$/+, Scr(YPWM*)$_{T1}$/+, Dll$_{T1}$ and Dll$_{T3}$ ChIP-seq signals, as well as ATAC$_{T1}$ and ATAC$_{T3}$ signals. Hox ChIP-seq peaks within the CRMs are numbered. The vertical gray bars denote the Hox ChIP peak center region that alters reporter expression when deleted. The *dsx-1* and *dpy-1* probes in Fig. 7 are also derived from the deleted regions. Panels on the right show T1 and T3 leg discs immunostained for reporter gene expression. The T1- and T3-specific expression patterns are indicated by arrows. Note that *dsx-1* drives expression only in male T1 leg discs, as expected for a *dsx* leg CRM. Scale bar: 100 μm.

We validated this assay by recapitulating Exd-facilitated Scr binding to two well-characterized Scr-Exd-binding motifs (Fig. 7f). Using this assay, Dll is also able to increase Scr binding to a Scr-Dll-binding motif derived from the SELEX data in a concentration-dependent manner. In contrast, a negative control protein, mCherry, showed no effect on the binding of Scr to this DNA sequence (Fig. 7g, h).

In summary, these results support the idea that Dll is a Hox cofactor, but that the mechanism by which Dll binds DNA with Scr is distinct from the highly cooperative binding exhibited by Hox–Exd heterodimers.

**A CRM from the *dsx* gene is activated by Scr-Dll in T1 leg discs.** To further test the role of Dll in contributing to paralog-specific Hox functions in vivo, the activities of putative CRMs bound by both Scr and Dll in our ChIP-seq datasets were assessed using *lacZ* reporter genes. Nine Exd-independent Scr$_{T1}$ > Ubx$_{T3}$ fragments co-occupied by Dll were tested. Seven generated T1 = T3 expression

patterns but the remaining two, named *dsx-1* and *dpy-1*, drove T1 ≠ T3 expression patterns in the Dll domain of leg discs (Fig. 8), consistent with them being direct paralog-specific Scr-Dll targets.

The *dsx-1* CRM, which is from the *doublesex* (*dsx*) gene, drives expression in two groups of cells at the center of T1 leg disc from males, with no expression observed in T3 leg discs or in female T1 leg discs (Fig. 8b). Both the expression pattern and the sexually dimorphic activity of this CRM agrees with the endogenous *dsx* expression, which is required for the development of sex combs bristles in the tarsal segments of male T1 legs[34]. A previous study[35] identified an early foreleg enhancer that overlaps with the *dsx-1* CRM, but does not recapitulate the endogenous *dsx* expression as faithfully as the *dsx-1* CRM. The *dpy-1* CRM drives expression in a crescent pattern in the Dll domain (Fig. 8c). Unlike *dsx-1*, the expression driven by *dpy-1* is specific to the T3 leg disc, potentially reflecting a role for repression by Scr.

The *dsx-1* CRM has three Scr ChIP-seq peaks, two of which (peaks 1 and 2) are Exd-independent and co-occupied by Dll

(Fig. 8b). Both of these peaks have multiple potential homeodomain binding motifs near the peak center, including several matches to our *NRLB*-derived Scr-Dll dimer motifs. Further, Dll occupancy at peak 2 is stronger in T1 compared to T3, consistent with the T1 > T3 expression pattern (Fig. 8b). Notably, the DNA sequence at the center of this peak shows an interesting phylogenetic pattern: the sequence is highly conserved among *Drosophila* species with sex combs, and is absent in species without sex combs (Supplementary Fig. 6). Due to the presence of multiple Scr-Dll motifs, we introduced small deletions at the peak center. While a short deletion of the sequences corresponding to peak 1 does not affect reporter expression, deleting 40 bp from peak 2 causes a delayed and weakened expression of the reporter gene, with the most obvious difference in young pupa (Fig. 8b). In vitro, Dll is able to enhance Scr binding to one of several Scr-Dll binding motifs located at the center of peak 2 (Fig. 7g, h).

For the *dpy-1* CRM, there are two Scr ChIP peaks, and both are Exd-independent. As with the sequence from *dsx-1*, Dll assisted Scr binding to the putative Scr-Dll binding motif from peak 2 (Fig. 7g, h). Further, although a small deletion at the center of peak 1 does not affect reporter expression, deleting about 40 bp from the center of peak 2 resulted in no expression in either T1 or T3 leg discs (Fig. 8c). These results suggest that this deletion removed an input for an essential transcription activator, precluding us from determining if Scr-Dll is a repressor of this CRM in the T1 leg disc. Alternatively, it is also possible that Scr, although bound to this CRM in vivo, does not regulate its activity.

Lastly, Fig. 5a shows that not all Exd-independent $Scr_{T1} > Ubx_{T3}$ CRMs have Dll co-occupancy, implying the existence of additional cofactors that facilitate paralog-specific binding. To test this, we generated reporter genes for six Exd-independent $Scr_{T1} > Ubx_{T3}$ fragments without Dll binding and, of these, three displayed T1 > T3 expression patterns (Fig. 8a and Supplementary Fig. 7). Thus, we conclude that there are additional mechanisms and/or cofactors beyond Dll and Exd that contribute to paralog-specific Hox binding and activity.

## Discussion

In this study, we used a combination of whole-genome and mechanistic approaches to understand how serially homologous appendages, such as the fly T1 and T3 legs, obtain their unique morphologies due to the activities of parallel Hox gene networks. The very similar transcriptomes in the three pairs of leg discs suggest that the different morphologies are largely a consequence of changing the expression patterns of the same sets of genes. By comparing the genome-wide DNA-binding profiles of the two relevant Hox paralogs, Scr and Ubx, in their native physiological contexts, we found hundreds of paralog-specific Hox targets, accounting for ~8% of all binding events for these two Hox proteins. Next, we showed that differences in chromatin accessibility and Hox monomer binding preferences are unlikely to account for paralog-specific binding. Instead, we demonstrate that interaction with the Hox cofactor Exd explains a large fraction of Scr's paralog-specific binding events. Finally, we identified Dll as a Hox cofactor in the complementary distal domain of the leg disc. Results from RNA-seq, CBP ChIP, and reporter assays suggest that about 1/3 of the paralog-specific Scr-binding events are functional and lead to tissue-specific gene regulation. Thus, paralog-specific Hox–DNA binding, which is mediated by multiple cofactors including Exd and Dll, contribute significantly to paralog-specific Hox gene networks.

**Exd plays a major role in regulating paralog-specific Hox gene networks**. Previous in vitro studies provided compelling evidence that the DNA-binding specificities of different Hox–Exd dimers

are more divergent from each other than those of Hox monomers, a phenomenon termed latent specificity[8]. There have also been several in vivo examples in which paralog-specific Hox–DNA binding and target regulation was shown to depend on an interaction with Exd[36–40]. Here we show that, on a genome-wide scale, the interaction with Exd explains a significant fraction of paralog-specific Hox binding, which often leads to paralog-specific gene regulation.

Earlier work also suggested that there is a tradeoff between specificity and affinity for Hox–Exd-binding motifs, where high affinity binding motifs are more likely to have low specificity for different Hox–Exd heterodimers[39]. We find that the paralog-specific, Exd-dependent CRMs characterized here (*ac-1*, *h-1*, and *fj-1*), have higher affinity Hox–Exd-binding motifs than those previously described in the *shavenbaby* (*svb*) gene:[18] the major Scr-Exd motif in the *fj-1* CRM has an affinity of about 0.06 relative to the optimal motif in the genome, while the motifs in *ac-1* and *h-1* have even higher relative affinities of nearly 0.15 and 0.2, respectively (Fig. 6b–d). In contrast, the Ubx-Exd-binding motifs in CRMs from *svb* have a relative affinity of <0.01[18]. One possible explanation for this difference is that the *svb* CRMs are active in embryos, which have many different cell types, while the CRMs characterized here are active in leg discs, which have significantly less cell-type complexity. Embryonic CRMs may require especially low-affinity binding motifs to distinguish their activities in a context with many cell types. Consistent with this idea, the *fkh250* CRM, which is also active in embryos, uses an Scr-Exd-binding motif with a low relative affinity of 0.017[18,36]. Notably, the relative affinities for the Scr-Exd-binding motifs in *ac-1*, *h-1*, and *fj-1* are at least eightfold higher than for Ubx-Exd (Fig. 6b–d). Manual inspection of other intergenic and intronic loci with $Scr_{T1} > Ubx_{T3}$ binding suggests that there are many other CRMs that follow this same rule. Thus, for specificity to occur, the most relevant feature may be that the affinity for the "correct" TFs, in this case Scr-Exd, must be significantly greater compared to the affinity for other "incorrect" TFs that are co-expressed in the same or homologous cells.

**Dll, a Hox cofactor**. Because Exd is only nuclear in a subset of cells during *Drosophila* development, such as the proximal domain of the leg disc, it was unlikely that Exd was the only Hox cofactor. In fact, CRMs that are directly regulated by Ubx have been described in cells where Exd is not available to be a cofactor[41,42]. However, it has remained an unresolved question whether non-Exd cofactors are used in these examples. More generally for the leg imaginal disc, the entire distal domain, extending from the trochanter to the tarsus, is without nuclear Exd, yet has Hox-dependent segment-specific morphological characteristics, such as the sex combs on the male T1 leg. Although several candidate TFs have been proposed to be Hox cofactors[27–30], none have been confirmed. In this study, we provide evidence that Dll is a distally acting Hox cofactor in leg discs.

There are many differences between how Exd and Dll interact with Hox proteins when bound to DNA. The Scr-Exd-binding motif is comprised of two partially overlapping half-sites, while the Scr-Dll motif consists of two HD-binding motifs separated by a spacer of several base pairs. Another difference is that the amount of cooperativity observed for Hox–Exd is far greater than that observed for Hox–Dll. The overlapping nature of the Hox- and Exd-binding motifs may be important for latent specificity, which for Scr requires an Exd-induced conformational change of the homeodomain[43]. In contrast, there is no evidence that latent specificity occurs as a consequence of Hox–Dll binding. Instead, the modest cooperativity observed for the Scr-Dll heterodimer is

likely a consequence of increasing Scr–DNA-binding affinity via a protein–protein interaction and closely spaced Dll and Scr-binding motifs.

More generally, we suggest that mode of DNA binding exhibited by Hox–Exd, which is highly cooperative and reveals latent specificity, may be the exception rather than the rule for TF–TF interactions within CRMs, and that the Scr-Dll example, with weak cooperativity between TFs stemming from a protein–protein interaction, may be the more common mode of interaction to distinguish the binding of paralogous TFs. In support of this notion, a systematic in vitro study identified 315 TF–TF interactions, only five of which exhibited latent specificity[44].

The TALE homeodomain proteins, which include Exd and Hth, are very ancient TFs that were present before the split of plants and animals, and TALE-mediated nuclear localization analogous to the Hth-Exd example in flies has been described in plants[45]. In contrast, the Hox gene family is only present in metazoans[46], and Dll is specific to bilaterians[47]. Moreover, it has been proposed that Dll initially functioned in the CNS, and was later co-opted to pattern the distal appendage[47]. Based on these observations, it is plausible that the Hox–Dll interaction evolved more recently than the Hox–Exd interaction, accounting for why Exd interacts with all Hox paralogs, while Dll may be a more limited Hox cofactor. This is supported by the results from a small-scale bimolecular fluorescence complementation (BiFC) screen that revealed Dll interacts with some Hox proteins, but not others[28].

Notably, the combined activities of Exd and Dll still do not account for all $Scr_{T1} > Ubx_{T3}$-binding events genome-wide and our reporter analysis suggests the presence of additional, yet to be identified Hox cofactors that have the capacity to promote Scr-specific binding. We suggest that the Hox–Dll mode of binding uncovered here may be representative of additional TFs that also have the ability to promote paralog-specific Hox binding and activity at specific CRMs. Further, we note that the differentiation of the T1 and T3 leg fates is a continuous developmental process and that the observations described here are limited to the late 3rd instar stage. Nevertheless, we expect that the principles governing Hox paralog specificity uncovered here will likely extend to other developmental stages and tissues. Finally, although we focus here on the role of paralog-specific TF-DNA binding, we note that there may be additional mechanisms that do not depend on differences in DNA binding between paralogous TFs that also contribute to their specific functions.

## Methods

**Testing a pair of TALENs targeting the *Scr* locus.** A pair of TALENs targeting the sequence between the ATG start codon and the YPWM encoding sequence of the *Scr* gene were purchased from the University of Utah Mutation Generation and Detection Core Facility. To make sure there were no SNPs relative to the reference *Drosophila melanogaster* genomic sequence that might interfere with the TALENs, the genomic fragment near the desired TALEN target site was PCR-amplified from the *yw* strain and the *ligase 4* mutant strain (Bloomington #28877), two possible recipient strains for the TALEN targeting experiments, and sequenced. The actual *Scr* locus genomic sequence was analyzed by the University of Utah Mutation Generation and Detection Core Facility, and a few satisfactory candidate TALEN targets were identified. Eventually, one target was chosen, and two plasmids encoding the TALEN pair targeting the chosen locus were generated. The TALEN target sequence and the sequences of the TALEN encoding plasmids are in Supplementary Information.

The TALEN encoding plasmids were linearized with NotI (NEB R0189S), which cut once downstream of the TALEN ORF, and gel purified. The linearized plasmids were used as templates to generate mRNA in vitro transcription using the AmpliScribe SP6 Transcription Kit (Epicentre AS3106), followed by capping using the ScriptCap m7G Capping System (Cellscript C-SCCE0625). A mix containing 200 ng/μl of each TALEN mRNA was used to inject the *yw* strain to test the efficiency of the TALENs. The injections were performed by BestGene Inc.

The injected G0 flies were individually crossed to *MKRS/TM6B* flies, and the F1 males were screened for reduced number of sex comb teeth, the classic *Scr*

phenotype. About 15% of G0 flies gave at least one male F1 with this *Scr* loss-of-function phenotype. Stocks were generated from a few selected F1 males with the *Scr* phenotype and analyzed. All had frameshift mutations (most of them were deletions, but a few were insertions) at the TALEN target site, and failed to complement with classic *Scr* null alleles $Scr^2$ and $Scr^4$. One of such alleles, named $Scr^{C8-1}$, has a 47 bp deletion at the TALEN site, and is predicted to encode a 32 amino acid peptide, and only the first 10 amino acids match the wild-type Scr peptide sequence. This allele was used in a number of experiments in this study as the *Scr* null allele, and its full sequence is listed in Supplementary Information.

**The generation of *Scr* targeting donor plasmid.** The entire 8 kb *Scr* fragment containing all desired mutations was assembled from three smaller fragments: Scr-1, Scr-2, and Scr-3. Molecular cloning was performed using standard procedures, and all PCR reactions were performed with the Phusion DNA polymerase (NEB M0530S). All restriction enzymes were purchased from NEB, and all primer sequences are listed in Supplementary Data 3.

From genomic DNA extracted from the *ligase 4* mutant line (Bloomington #28877), the 3.6 kb Scr-1, 1.8 kb Scr-2, and 3.3 kb Scr-3 fragments were PCR-amplified using primers Scr-1-5' + Scr-1-3', Scr-2-5' + Scr-2-3' and Scr-3-5' + Scr-3-3', respectively. The purified PCR fragments were digested with XbaI + XhoI, and individually cloned into the pBluecript vector digested with XbaI + XhoI, generating constructs PBS-Scr-1, PBS-Scr-2 and PBS-Scr-3. All constructs were verified by restriction digestion and sequencing.

The YPWM-AAAA mutation was introduced into the PBS-Scr-2 construct. The PBS-Scr-2 construct was PCR-amplified using primers Scr-YPWM-AAAA-5' and Scr-YPWM-AAAA-3', followed by DpnI digestion at 37 °C. The digested DNA was used to transform DH5α competent cells, and the transformants were analyzed by DNA sequencing to identify clones successfully mutated. Next, the mutagenesis of TALEN targeting site and the insertion of the 3×FLAG tag were achieved sequentially by overlapping extension PCR-based mutagenesis. In each round of mutagenesis, the plasmid was used as the template, and M13 primer + reverse mutagenesis primer, as well as M13R primer + forward mutagenesis primer, were used as primer combinations to PCR amplify the two half fragments. The two half fragments were then used as the templates and M13 + M13R primers were used to amplify the complete mutant fragment. The mutant fragment was then digested with XbaI + XhoI, and cloned into pBluescript vector digested with XbaI + XhoI. All constructs were verified by restriction digestion and DNA sequencing. The final construct was named as PBS-Scr-2(m).

Next, the Scr-1 fragment was excised from PBS-Scr-1 by XbaI + BclI digestion, and the mutant Scr-2 fragment was excised from PBS-Scr-2(m) by BclI + RsrII digestion. Both fragments were inserted into XbaI + RsrII digested PBS-Scr-3 through multi-fragment ligation, resulting in the construct PBS-Scr(m). This construct was verified by restriction digestion, and all ligation junctions were sequenced to make sure no mutations were introduced. A construct containing the 3xP3-RFP cassette flanked by multiple unique restriction sites was previously generated. In the last step of cloning, the 3xP3-RFP fragment was excised from this construct by ZraI + XhoI digestion, and ligated into the PBS-Scr(m) construct sequentially treated with AsiSI digestion, T4 DNA polymerase treatment to convert sticky ends to blunt ends, and second digestion with XhoI. The final targeting plasmid was verified by restriction digestion, and its ligation junctions were sequenced to make sure no mutations were introduced.

**The generation of 3×FLAG-Scr, 3×FLAG-Scr(YPWM-AAAA), and 3×FLAG-Ubx alleles.** Having verified the efficiency of the *Scr* TALENs, a mixture containing 500 ng/μl of each TALEN mRNA, as well as 500 ng/μl of the donor plasmid, was used to inject embryos. The *ligase 4* mutant line (Bloomington #28877) was selected as the recipient strain to suppress unwanted non-homologous end joining (NHEJ) events[48,49], therefore boosting the desired homologous recombination events. The injection was performed by BestGene Inc.

The injected G0 flies were individually crossed to *TM3/TM6B* flies, and in the next generation, paired crosses were set up between one single F1 male and one single F1 female from the same G0 cross. As many as 20 paired crosses were established for each G0 cross. Therefore, as many as 40 F1 flies from each G0 parent were screened. After a few days when F2 larval activity became obvious, the 20 F1 flies from 10 paired crosses were pooled in one 1.5 ml tube, and their genomic DNA extracted. PCR with *taq* DNA polymerase (NEB M0273S) followed by agarose gel electrophoresis was used to screen for the presence of the 3×FLAG tag and the YPWM-AAAA mutation. Primers 3×FLAG-check-5' + 3×FLAG-check-3' and YPWM-AAAA-check-5' + YPWM-AAAA-check-3' were used. If a positive signal was detected, the ten paired crosses that comprised the sample with the positive signal were then analyzed. A few F2 individuals from each paired cross were used to extract genomic DNA, and the same PCRs were used to screen for the positive signals. Once the signals were narrowed down to individual paired crosses, PCR was used to look for the presence of the 3xP3-RFP cassette using primers 3xP3-RFP-check-5' and 3xP3-RFP-check-3'. The presence of this cassette indicated whole plasmid integration events, and such stocks were excluded from further analysis. A few TM3 or TM6B balanced F2 males (each one might or might not have the desired mutation(s)) were selected from each positive paired cross to set up individual crosses and establish stocks. The final stocks were screened by PCR similarly for the presence of the 3×FLAG tag and/or the YPWM-AAAA mutation,

as well as the 3xP3-RFP cassette. Only one positive stock was kept from each G0 fly to make sure all lines were independent. Sometimes the same G0 gave both *3×FLAG-Scr* and *3×FLAG-Scr(YPWM\*)* alleles. In such cases, one stock of each genotype was kept. All final stocks were also screened under the fluorescent scope to make sure there was no eye-specific RFP expression, and were verified by southern blot analysis and DNA sequencing. The generation of the *3×FLAG-Ubx* allele was described in ref. [13]. *3×FLAG-Scr^{C18-6}*, *3×FLAG-Ubx*[7] and *3×FLAG-Scr(YPWM-AAAA)^{D8-16}* alleles were used throughout this study. The sequences of all primers used in the screening are listed in Supplementary Data 3.

**Southern blot**. Southern blot analysis was performed using the DIG High Prime DNA Labeling and Detection Starter Kit II (Roche 11585614910) and the DIG Wash and Block Buffer Set (Roche 11585762001). In all, 5–10 μg of genomic DNA (roughly genomic DNA extracted from about 15 adult flies) was digested with selected restriction enzymes in 30-μl reactions, and the entire samples were run on a 1% agarose gel. The DIG-labeled DNA marker II (Roche 11218590910) was used to determine band size, and appropriate amount of ClaI-digested targeting donor plasmid was used as a positive control. All subsequent treatments of the gel were performed at room temperature. After separating the DNA fragments by gel electrophoresis, the gel was denatured by two washes with 2.5 gel volumes of denature solution (0.5 M NaOH, 1.5 M NaCl), 15 min each, then neutralized by two washes with 2.5 gel volumes of neutralization solution (0.5 M Tris-HCl, 1.5 M NaCl, adjusted to pH 7.5 with HCl), 15 min each. The gel was then washed once with 2.5 gel volumes of 20×SSC (3 M NaCl, 300 mM sodium citrate, adjusted to pH 7.0 with HCl) for 10 min. A standard DNA transfer apparatus was then assembled, and the DNA on gel was transferred to Nylon membrane, positively charged (Roche 11417240001) for 20–24 h by capillary effect. 20×SSC was used as the transfer solution.

The DNA was then UV cross-linked to the Nylon membrane using a UV Stratalinker 1800 with build-in auto-cross-link settings. The membrane was then briefly washed with 2×SSC and prehybridized with 20 ml of DIG Easy Hyb at 42 °C for 1 h according to the manufacturer's instructions. The DIG-labeled probe was generated according to the manufacturer's instructions. The 8 kb *Scr* locus DNA fragment in the targeting donor plasmid was used as a template to generate a labeled probe. This fragment was too long so it was first digested with XmnI. In all, 800 ng of the digested DNA was used as a template and a 7-h labeling reaction was performed. After prehybridization, 6 μl of the DIG-labeled probe was added to 6 ml of DIG Easy Hyb, and hybridization was performed at 42 °C overnight.

The membrane was washed twice with 100 ml of 2×SSC, 0.1% SDS at room temperature for 15 min with gentle shaking. The membrane was then washed twice with 100 ml of 0.5×SSC, 0.1% SDS at 65 °C for 15 min with rotation in a hybridization oven, and the solution was pre-heated to 65 °C before the washes. All subsequent treatments of the membrane were performed at room temperature. The membrane was briefly rinsed with 30–50 ml of washing buffer, and blocked with 100 ml of blocking solution for 30 min with gentle shaking. Next, the membrane was incubated with 30 ml of antibody solution for 30 min with gentle shaking. The membrane was then rinsed with 30–50 ml of washing buffer, and washed twice with 100 ml of washing buffer for 15 min with gentle shaking. Finally, the membrane was equilibrated with 35 ml of detection buffer for 5 min. In total, 1 ml of the chemiluminescent substrate CSPD was applied to the membrane and incubated at 37 °C for 10 min. A standard film exposing cassette was assembled and incubated at 37 °C for 10 min. Films were then exposed for a desired time period to obtain optimal signal intensity.

**Imaging of adult flies and adult fly legs**. Legs from adult males of the isogenic *w^{1118}* line (Bloomington #5905) were imaged for Fig. 1a. The same isogenic *w^{1118}* line was also used in Fig. 3e (left) as the wild-type. In order to obtain adult flies consisted mostly of homozygous cells mutant for desired *Scr* alleles in thoracic appendages, the *Minute* technique[50] was used. *Dll-Gal4*, which is expressed in all leg disc cells early in development, is described in ref. [50], and the *Minute* allele *Rps3\** (Bloomington #5699) and *Ubi-mRFPnls* (Bloomington #30555) were ordered from the Bloomington Stock Center. Adults in Fig. 3e (middle) were obtained by the following cross: *Dll-Gal4, UAS-FLP/CyO, Act-GFP; FRT82B, Scr^{C8-1}/TM6B ⊗ FRT82B, Rps3\*, Ubi-mRFPnls/TM6B*, and adults in Fig. 4e (right) were obtained similarly by this cross: *Dll-Gal4, UAS-FLP/CyO, Act-GFP; FRT82B, 3×FLAG-Scr(YPWM-AAAA)^{D8−16}/TM6B ⊗ FRT82B, Rps3\*, Ubi-mRFPnls/TM6B*.

All fly legs and adult flies were imaged using Nikon SMZ18 stereomicroscope and processed with the Nikon software NIS-Elements D4.60.00.

**Counting sex comb teeth numbers**. For each genotype of interest, 40 T1 legs from 20 males were counted. The flies were rinsed in 100% ethanol to remove body wax, and then washed briefly with PBS + 0.1% Triton X-100. The T1 legs were removed from the flies and transferred to a slide with a drop of PBS + 0.1% Triton X-100. After all legs were transferred to the slide, the legs were adjusted under the dissection scope such that the sex comb teeth were all facing up, and all legs were aligned in two to three rows. A coverslip was placed on the samples, and the slide was sealed with nail polish. The number of sex comb teeth were then counted under a Zeiss Axio Imager microscope. The plots showing the final results were generated using the R package ggplot2.

**The generation of lacZ reporter flies**. The lacZ reporter constructs were generated by cloning PCR-amplified enhancer fragments into the lacZ vector pRVV54[51], which has an attB site, a mini-white marker gene, and a multiple cloning site upstream of the nuclear lacZ sequence. The genomic coordinates of each enhancer are detailed in Supplementary Data 4, and all primers used are listed in Supplementary Data 3, and all restriction enzymes were purchased from NEB.

All candidate enhancers were amplified by PCR from genomic DNA extracted from the isogenic *w^{1118}* line (Bloomington #5905) using the Phusion DNA polymerase (NEB M0530S). The PCR products were cloned into the pRVV54 vector by restriction cloning. All constructs were verified by restriction digestion and sequencing with pRVV54-up and pRVV54-down primers. For long enhancers, internal sequencing primers were also used. The PCR products were also sequenced with the PCR primers (and internal primers when applicable) to make sure there was no PCR-introduced mutations.

To mutate the selected Hox–Exd motifs or to generate small deletions, overlapping extension PCR was performed using the wild-type PCR product as a template. The forward primer and the reverse mutagenesis primer were used to amplify the left half fragment, and the reverse primer and forward mutagenesis primer were used to amplify the right half fragment. Then both half fragments were used as templates, and the two PCR primers were used to perform overlapping extension PCR. The final mutant PCR products were digested with the same restriction enzymes selected for the corresponding wild-type PCR products, and cloned into the pRVV54 vector. All mutant reporter constructs were verified by restriction digestion and sequencing.

All verified reporter constructs were injected into recipient flies with the attP40 landing site, and transformants were selected by the presence of the mini-white marker. All injections were performed by BestGene Inc.

**Leg disc antibody staining**. The isogenic *w^{1118}* line (Bloomington stock #5905) was used in the staining in Fig. 1b, the yw line was used in Figs. 3f and Fig. 5c, and fly lines bearing lacZ reporter transgenes were stained with β-Gal antibody. The following primary antibodies were used in this study: polyclonal guinea pig anti-Scr (GP111) was a custom-made antibody and was used at 1:2000 (Fig. 1a), monoclonal mouse anti-Scr (6H4.1, hybridoma bank) was used at 1:40 (Fig. 5c), monoclonal mouse anti-Ubx (FP3.38, hybridoma bank, ascites) was used at 1:100, monoclonal mouse anti-Ac (hybridoma bank) was used at 1:5, polyclonal rabbit anti-β-Gal (MP Biomedicals, cat #559762, lot #06825) was used at 1:4000, polyclonal guinea pig anti-Hth[36] was used at 1:2000 (Fig. 3f), polyclonal rabbit anti-Hth[52] was used at 1:1000 (Fig. 5c) and guinea pig anti-Dll[53] was used at 1:2000. The following commercial secondary antibodies were used in this study: goat anti-mouse IgG Alexa Fluor 488 (Molecular Probes A11029), Goat anti-rabbit IgG Alexa Fluor 488 (Molecular Probes A11034), Goat anti-GP IgG Alexa Fluor 488 (Molecular Probes A11073), goat anti-guinea pig IgG Alexa Fluor 647 (Molecular Probes A-21450), Goat anti-mouse IgG Alexa Fluor 555 (Molecular Probes A-21424), goat anti-guinea pig IgG Alexa Fluor 555 (Molecular Probes A-21435), and goat anti-rabbit IgG Alexa Fluor 555 (Molecular Probes A-21429). All secondary antibodies were used at 1:1000 except goat anti-guinea pig IgG Alexa Fluor 647, which was used at 1:500.

Leg disc antibody staining was performed using standard protocol. Briefly, wandering larvae of desired genotype were pulled apart, and the anterior halves were inverted in PBS. The gut, fat bodies, and salivary glands were then removed, followed by fixation in PBS + 0.1%Triton X-100 + 4% formaldehyde at room temperature with rotation for 20 min. After washing 3 times with PBS + 1% Triton X-100 at room temperature with rotation, 5 min each, the samples were blocked with blocking solution (PBS + 1% Triton X-100 + 1% BSA) for 1 h at room temperature with rotation. The samples were then incubated with the primary antibody in blocking solution at 4 °C overnight with rotation. Next, the samples were rinsed briefly with PBS + 1% Triton X-100, followed by three washes with PBS + 1% Triton X-100 at room temperature with rotation, 30 min each. The samples were then incubated with secondary antibody in blocking solutions for 2 to 4 h in the dark at room temperature with rotation. Next, the samples were briefly rinsed with PBS + 1% Triton X-100, and then washed three times with PBS + 1% Triton X-100 in dark at room temperature with rotation, 30 min each. The target discs were then dissected from the samples in PBS + 1% Triton X-100, and transferred to a 1.5-ml tube containing PBS + 1% Triton X-100. The supernatant was removed and a drop of Vectashield mounting medium with DAPI (Vector Laboratories H-1200) was added. The samples were placed at 4 °C overnight in dark to let the discs settle. The discs were then mounted on a slide and imaged with Leica SP5 II confocal microscope. The images were processed with the software Fiji. For each staining, at least ten different samples were investigated, and the same conclusion could be reached.

**Embryo antibody staining**. The *yw* line was used as wild type. The null allele *Scr^{C8-1}* and the YPWM motif mutant allele *3×FLAG-Scr(YPWM\*)^{D8−16}* are both homozygous lethal, so they were balanced with the *TM3, twi-Gal4, 2xUAS-EGFP* balancer[54] in order to identify homozygous mutant embryos. The following primary antibodies were used: polyclonal rabbit anti-CrebA[55] was purchased from DSHB and was used at 1:20,000 (after three rounds of pre-absorption with wild-type embryos to reduce non-specific signal), and chicken anti-GFP (Abcam ab13970) was used at 1:1000. The following secondary antibodies were used, all at

1:1000 dilution: goat anti-rabbit IgG Alexa Fluor 555 (Molecular Probes A-21429), and goat anti-chicken IgY DyLight 488 (Invitrogen SA5-10070).

The embryos were collected and stained with standard protocols. Briefly, the embryos were collected overnight using standard embryo collection cages supplied with fresh yeast paste. The embryos were dechorionated with 50% bleach at room temperature for 3 min, followed by thorough washes with deionized water. The dechorionated embryos were then transferred into a glass vial containing 1 volume of heptane and 1 volume of PBS + 4% formaldehyde, and shaken vigorously for 20 min at room temperature. The lower phase, as well as any embryos in it, was removed, and 1 volume of methanol was added. The vial was shaken vigorously for 1 min at room temperature to remove the vitelline membrane. The devitellinized embryos should sink to the bottom of the glass vial, and were collected and transferred to 1.5-ml tubes. The fixed embryos were washed three times with methanol, and could be stored at −20 °C for months before antibody staining.

The fixed embryos were rehydrated by washing once with methanol, once with 1:1 mix of methanol and PBS + 1% Triton X-100, and twice with PBS + 1% Triton X-100. All washes were 5 min each at room temperature with rotation. The embryos were then blocked in blocking solution (PBS + 1% Triton X-100 + 1% BSA) at room temperature for 1 h with rotation. The blocked embryos were incubated with primary antibody in blocking solution at 4 °C overnight. Next, the embryos were briefly rinsed and then washed three times with PBS + 1% Triton X-100 at room temperature with rotation, 30 min each. The embryos were then incubated with secondary antibody in blocking solution for 2–4 h in the dark at room temperature with rotation. Next, the embryos were rinsed briefly with PBS + 1% Triton X-100, followed by three 30-min washes with PBS + 1% Triton X-100 in the dark at room temperature with rotation. The supernatant was removed from the tubes, and a few drops of Vectashield mounting medium with DAPI (Vector Laboratories H-1200) were added to each sample. The stained embryos were stored at 4 °C overnight to let the embryos settle to the bottom of the tubes. The embryos were then mounted on slides and imaged using a Leica SP5 II confocal microscope. All images were processed with the Fiji software. For each staining, at least ten different samples were investigated, and the same conclusion could be reached.

**Preparation of disc chromatin for ChIP.** The procedure for preparing chromatin from imaginal discs was modified from a previously published protocol[53]. About 200 leg discs were used in each ChIP experiment. Wandering larvae of the desired genotype were taken from vials and washed thoroughly to remove any food debris. The larvae were pulled apart in room temperature PBS, and the anterior halves were immediately transferred to ice-cold PBS. After all samples were transferred, they were inverted in ice-cold PBS, and the samples were kept cold as much as possible during the procedure. The inverted samples were then cross-linked in a 15-ml falcon tube containing 10 ml of crosslinking solution (10 mM HEPES pH 8.0, 100 mM NaCl, 1 mM EDTA pH 8.0, 0.5 mM EGTA pH 8.0, filtered) plus freshly added 270 μl of 37% formaldehyde (final formaldehyde concentration ≈1%). The samples were rotated in room temperature for 10 min. Next, 1 ml of 2.5 M glycine was added, and the tube was inverted by hand for about 1 min. After the samples were settled to the bottom of the tube, the supernatant was removed with a pipette, and 10 ml of quench solution (1×PBS, 125 mM glycine, 0.1% Triton X-100, autoclaved) was added and the tube was rotated at room temperature for at least 6 min. The samples were then washed twice with 10 ml of ice-cold buffer A (10 mM HEPES pH 8.0, 10 mM EDTA pH 8.0, 0.5 mM EGTA pH 8.0, 0.25% Triton X-100, filtered) plus protease inhibitors (cOmplete™, Mini, EDTA-free Protease Inhibitor Cocktail, Roche 11836170001), 10 min each at 4 °C with rotation. Next, the gut, fat bodies, and salivary glands were removed from all samples in ice-cold buffer A with protease inhibitors, and the samples were kept cold as much as possible. The cleaned samples were then washed twice with 10 ml of ice-cold buffer B (10 mM HEPES pH 8.0, 200 mM NaCl, 1 mM EDTA pH 8.0, 0.5 mM EGTA pH 8.0, 0.01% Triton X-100, filtered) plus protease inhibitors (same as in buffer A), 10 min each at 4 °C with rotation. The target leg discs were then removed from the samples in ice-cold buffer B with protease inhibitors and were transferred to a 1.5-ml tube placed on ice containing 0.5 ml of ice-cold buffer B with protease inhibitors. The samples were again kept cold as much as possible. Once all discs were dissected, they were transferred to a 15-ml falcon tube. After the discs settle to the bottom, the supernatant was removed, and 0.9 ml of buffer C (10 mM HEPES pH 8.0, 1 mM EDTA pH 8.0, 0.5 mM EGTA pH 8.0, 1% Triton X-100, filtered) plus protease inhibitors (Halt™ Protease Inhibitor Cocktail, EDTA-Free (100X), Thermo Fisher 87785) was added. Next, the samples were sonicated in the 15-ml falcon tube in ice water bath using Branson Sonifier 450, at 15% amplitude for 12 min, 15 s on/30 s off. After sonication, the 15-ml tube was briefly spun to collect all samples to the bottom, and the entire sample was transferred to a 1.5-ml tube. The chromatin sample was then centrifuged in a refrigerated tabletop centrifuge at the max speed at 4 °C for 10 min to remove any insoluble materials, and 850 μl of the supernatant was transferred to a new 1.5-ml tube. The chromatin may be used immediately for ChIP, or could be flash-frozen in liquid nitrogen, and stored at −80 °C for at least a few weeks. Generally, about 1 to 1.5 μg of chromatin could be expected from about 200 leg discs.

**ChIP.** The following antibodies were used in the ChIP experiments performed in this study. Monoclonal mouse anti-FLAG M2 (Sigma F1804, 10 μg per ChIP),

guinea pig anti-Dll[53] (5 μl per ChIP), goat anti-Hth (Santa Cruz sc-26187, lot A1204, 3 μg per ChIP) and rabbit anti-CBP (a gift from Mattias Mannervik, 5 μl per ChIP, preabsorbed using wild-type embryos). Normal mouse IgG (Santa Cruz Biotechnology, sc-2025), normal guinea pig IgG (Santa Cruz Biotechnology, sc-2711), and normal rabbit IgG (Santa Cruz Biotechnology, sc-2027, or Thermo Fisher Scientific, 10500 C) were used in pre-clearing of the samples.

The ChIP protocol used in this study was derived from two previously published procedures[56,57]. All buffers were pre-chilled on ice before use, and the samples were kept cold for as much as possible during all handling steps.

Day 1: 1/4 volume of 5× chromatin dilution buffer (50 mM Tris-HCl pH 8.0, 5 mM EDTA pH 8.0, 750 mM NaCl, 1% Triton X-100, filtered) was added to each chromatin sample to adjust buffer condition, and appropriate volume of Halt™ Protease Inhibitor Cocktail, EDTA-Free (100×) was then added to each sample. Next, 10 μg of normal IgG from the same host species as the ChIP antibody was added to each chromatin sample to pre-clear the samples. The sample was then rotated at 4 °C for 1 h. In the meantime, the protein G agarose beads (Roche 11243233001) were prepared. In total, 40 μl of the beads suspension (settled beads volume 20 μl) were used for each ChIP reaction, and the same amount was used for each pre-clearing reaction. The appropriate volume of beads was washed twice with 1 ml of RIPA buffer (10 mM Tris-HCl pH 8.0, 1 mM EDTA pH 8.0, 150 mM NaCl, 1% Triton X-100, filtered), 10 min each at 4 °C with rotation. To each aliquot of beads for ChIP reactions, add 12.5 μl of 100 mg/ml BSA (Sigma A2153) and 25 μl of 10 mg/ml tRNA (Roche 10109517001), and rotate at 4 °C overnight to block. Add the chromatin samples to the aliquots of beads for pre-clearing, and rotate at 4 °C for 1 h. The samples were then spun at the max speed at 4 °C for 10 min. Most of the supernatant was transferred to new 1.5-ml tubes, and about 130 μl was left in the old tube. To each precleared chromatin sample in the new tube, add 12.5 μl of 100 mg/ml BSA, 25 μl of 10 mg/ml tRNA, and appropriate amount of ChIP antibody, and rotate at 4 °C overnight. From each leftover chromatin sample in the old tube, take 100 μl and store at −80 °C as input.

Day 2: The blocked beads were separated from the supernatant by spinning, and the supernatant was discarded. The chromatin samples were added to the beads, and the samples were then rotated at 4 °C for 3 h. Next, the beads were rinsed with 1 ml of RIPA buffer, followed by two washes with RIPA buffer, one wash with high salt RIPA buffer (10 mM Tris-HCl pH 8.0, 1 mM EDTA pH 8.0, 350 mM NaCl, 1% Triton X-100, filtered), one wash with LiCl buffer (10 mM Tris-HCl pH 8.0, 1 mM EDTA pH 8.0, 250 mM LiCl, 0.1% IGEPAL CA-630, filtered), and 1 wash with TE buffer (10 mM Tris-HCl, 1 mM EDTA, pH 8.0, filtered). Each wash was done with 1 ml of buffer at 4 °C with rotation for 10 min. After the TE wash, resuspend the beads with 500 μl of TE. The input samples were also adjusted to 500 μl with TE buffer. Next, to all beads and input samples, add 5 μl of 5 M NaCl, 12.5 μl of 20% SDS, and 10 μl of 1 mg/ml RNase (Sigma R5503), and incubate at 37 °C for 30 min with rotation. Overall, 20 μl of 20 mg/ml proteinase K (Roche 03115836001) was then added to each sample. The samples were incubated at 55 °C for 2 to 3 h with rotation, followed by rotation at 65 °C overnight to decrosslink. The 37 °C, 55 °C, and 65 °C incubation steps were all done in a hybridization oven, and to avoid leaking of samples, DNA LoBind Safe lock tubes (Eppendorf 022431021) were used.

Day 3: The ChIP samples were centrifuged at max speed at room temperature for 1 min, and the supernatant was transferred to new tubes. In all, 100 μl of 3 M sodium acetate (pH 5.2) was added to each sample (ChIP and input), and the samples were extracted with phenol:chloroform (1:1) and then with chloroform. In total, 1 μl of 20 mg/ml glycogen (Roche 10901393001) was then added to each sample, and DNA in the samples was purified by isopropanol precipitation. The DNA pellet was dissolved with 30 μl of 10 mM Tris buffer, pH 8.0. Finally, 5 μl of each ChIP sample and 5 μl of each 1:10 input sample were used to quantify the amount of purified DNA using Qubit fluorometer with Qubit dsDNA HS Assay Kit (Thermo Fisher Scientific Q32854). After sacrificing some input and ChIP samples for quantification, the amount of DNA left for library preparation was generally the following: about 10 ng for input samples and about 1.5 ng for ChIP samples.

**ChIP-seq library preparation.** ChIP-seq libraries were prepared using the TruSeq ChIP Library Preparation Kits (Illumina IP-202-1012 and IP-202-1024), following the manufacturer's instructions. About 8–10 ng of DNA (or the entire samples if there was less than this amount for some ChIP samples) was used as the starting materials, 16 cycles of PCR amplification were performed for all libraries. The libraries were first quantified using nanodrop, and appropriate dilutions were made for accurate quantification and size determination. The libraries were then quantified with Qubit fluorometer with Qubit dsDNA HS Assay Kit (Thermo Fisher Scientific Q32854), and the library sizes were determined by bioanalyzer, using Bioanalyzer High Sensitivity DNA Analysis (Agilent 5067-4626).

**ATAC-seq library preparation.** The ATAC-seq library preparation procedure was modified from ref. [17]. *3×FLAG-Scr* wondering larvae were dissected in PBS + 1% BSA on ice for T1 or T3 leg discs. BSA was added to prevent the discs from sticking to plasticware. The discs were resuspended in nuclear extraction buffer (NEB, 10 mM HEPES pH 7.5, 2.5 mM MgCl₂, 10 mM KCl) and placed in a 1 mL Dounce homogenizer (Wheaton 357538) on ice. The discs were homogenized with 15 strokes by the loose pestle, followed by a 10-min incubation on ice, then with 20 strokes by the tight pestle. The dissociated nuclei were counted using a

hemocytometer, and 50,000 nuclei were transferred to a 1.5-ml tube containing 1 mL of NEB + 0.1% Tween-20. The sample was briefly mixed, and then immediately spun at 1000×g at 4 °C for 10 min to pellet the nuclei. The transposition reaction was performed using the Nextera DNA Library Preparation Kit (Illumina FC-121-1030). The supernatant was removed and the pellet was resuspended in 50 μl of freshly prepared ATAC transposition solution (1×TD buffer (2×TD is supplied in the Illumina kit), 0.1% Tween-20, 0.01% digitonin, 1/20 volume of the Tn5 transposase (supplied in the Illumina kit)). The transposition reaction was performed on a thermomixer at 1000 rpm at 37 °C for 30 min, and the DNA was purified using the MinElute PCR Purification Kit (Qiagen 28006). The DNA was eluted with 2 × 11 μl of the elution buffer, and 20 μl of eluted DNA was used for PCR amplification. The number of PCR cycles was determined according to ref. [17]. Library DNA was size selected and purified using the AMPure XP beads (Beckman Coulter A63881). Two-sided size selection using 0.55 volume and 1.65 volumes of the beads was performed, and 21 μl of nuclease-free water was used to elute the library DNA.

**Generation of protein expression constructs.** pET9a-Exd expresses untagged full-length Exd and pET14b-Hth[HM] expresses the HM isoform of Hth with N-terminal 6xHis tag, and these vectors were described before[8]. The pET21a-T7-Dll-his vector expresses full-length Dll-PB isoform with N-terminal T7 tag and C terminal 6×His tag, and was described in ref. [58].

The following protein expression vectors were generated in this study: pQE30-EGFP, pQE30-mCherry, pQE30-EGFP-Scr-FLAG, and pET9a-Exd-T7. All cloning steps involving pQE30 backbone require the host cells to express high levels of the lacI protein, and were performed using 5-alpha F' Iq cells (NEB C2992H).

The EGFP fragment was amplified using primers EGFP-5' and EGFP-3', and the TEV-MCS fragment, which had KpnI and SalI overhangs, was generated by annealing oligos TEV-MCS-5' and TEV-MCS-3'. The TEV-MCS fragment and BamHI + KpnI digested EGFP fragment was ligated into BamHI + SalI-digested pQE30 vector (Qiagen) in a 3-fragment ligation reaction, generating pQE30-EGFP. The pQE30-mCherry construct was generated by replacing the BamHI-AvrII EGFP fragment by the BamHI-AvrII mCherry fragment, which was PCR-amplified using primers mCherry-5' and mCherry-3', followed by BamHI + AvrII digestion. The full-length Scr ORF was amplified using primers Scr-FL-5' and Scr-FL-3' (which has the FLAG encoding sequence), digested with SpeI + AscI, and ligated into SpeI + AscI-digested pQE30-EGFP to generate pQE30-EGFP-Scr-FLAG.

The Exd-T7 fragment was PCR-amplified using pET9a-Exd as the template and T7-promoter and Exd-T7-3' as the primers. This fragment was digested with NdeI + BamHI, and was used to replace the NdeI-BamHI Exd fragment of pET9a-Exd to generate pET9a-Exd-T7.

The pQE30-based expression constructs were used to transform the M15 E. coli cells (Qiagen) to generate the protein expression strains. pET9a-Exd-T7 and pET14b-Hth[HM] were used to co-transform BL21(DE3) cells to generate the strain that expresses Exd-T7 (with Hth[HM]).

**Recombinant protein expression and purification.** In all, 5 ml of LB medium with appropriate antibiotics was inoculated with the protein expression strain, and the culture was shaken at 37 °C overnight. In the next morning, 1 ml of overnight culture was used to inoculate 150 ml of fresh LB medium with appropriate antibiotics. The culture was shaken at 37 °C until OD600 reached about 0.7. IPTG was added to a final concentration of 1 mM, and the culture was shaken for another 5–6 h at 37 °C before harvesting the cells.

The cells were resuspended in 8 ml of Lysis/wash buffer (50 mM Tris pH 7.5, 500 mM NaCl, 20 mM Imidazole) with a proteinase inhibitor cocktail (Roche 11836170001), and sonicated to lyse the cells. The samples were centrifuged at 4 °C at 10,000 rpm (~12,000×g) for 30 min. The supernatant was loaded onto Ni-NTA agarose beads (Qiagen 30210) rinsed with lysis/wash buffer. Binding was performed at 4 °C for 2 h with rotation. The beads were washed three times with lysis/wash buffer, and each wash was performed at 4 °C for 5 min. Elution was performed at room temperature with 125 μl of elution buffer (lysis/wash buffer supplied with 300 mM imidazole and proteinase inhibitor cocktail) for 10 min, and the elution was repeated once. The eluates were pooled and dialyzed at 4 °C overnight with dialysis buffer (20 mM HEPES pH 8.0, 200 mM NaCl, 10% glycerol, 2 mM MgCl2) using Slide-A-Lyzer Dialysis cassette (Thermo Scientific 66383). 0.05% was included in all buffers when purifying Dll. The protein samples were quantified with Bradford assay (Biorad 500-0006) using BSA as the standard, and were analyzed on SDS-PAGE.

The following is a list of proteins used in the figure panels. Fig. 5e: EGFP-Scr-FLAG and T7-Dll. Fig. 7a–d: EGFP-Scr-FLAG, T7-Dll and Exd-T7 (with Hth[HM]). Fig. 7f: EGFP-Scr-FLAG and Exd-T7 (with Hth[HM]). Fig. 7h: EGFP-Scr-FLAG, T7-Dll, and mCherry.

**Gel-free SELEX.** To make the R0 SELEX library, 10 μl of 10x STE buffer (100 mM Tris pH 8.0, 10 mM EDTA pH 8.0, 1 M NaCl), 10 μl of 100 μM SELEX library oligo, 20 μl of 100 μM SELEX-R primer, and 60 μl H2O were mixed. The mixture was denatured by boiling for 10 min, and cooled to room temperature slowly to anneal the primer to the library oligo. Klenow reaction was used to generate the double-stranded library DNA. 25 μl of 10× NEBuffer 2, 20 μl of 10 mM dNTP,

80 μl H2O, and 25 μl Klenow fragment (NEB M0210L) were then added to the sample, and the sample was incubated at room temperature 30 min. In total, 10 μl of 0.5 M EDTA, pH 8.0 was used to stop the reaction, and the sample was divided into five parts and each part was purified using one PCR purification columns (PCR purification kit, Qiagen 28106). In total, 50 μl elution buffer was used the elute each column, and the eluates were pooled.

16mer R0 libraries were used for all monomer SELEX experiments, and 24mer libraries were used for all dimer SELEX experiments. The 50 μl SELEX reaction samples were assembled by mixing 25 μl of protein mixture and 25 μl of the DNA library mixture. The DNA library mixture contained 10 μl of 5x binding buffer (50 mM Tris-HCl pH 7.5, 250 mM NaCl, 5 mM MgCl2, 20% glycerol, 2.5 mM DTT, 2.5 mM EDTA, ~125 ng/μl polydIdC (Sigma P4929-10UN), 100 ng/μl BSA), 7.5 μl H2O, and 7.5 μl of 3.3 μM dsDNA library, and the protein mixture was generated by mixing appropriate volumes of 1 μM proteins and adjusted to 25 μl with dialysis buffer. 0.05% Tween-20 was included when Dll was used in the reactions. The DNA library mixture and the protein mixture were combined and incubate at room temperature for 30 min. In the final 50 μl sample, the dsDNA library was at a concentration of 500 nM, and the protein concentrations were: 100 nM for Scr and Ubx, 50 nM for Dll and 200 nM for Exd (with Hth[HM]).

In all, 30 μl of Dynabeads™ Protein G (Thermo Fisher 10004D) was rinsed once with 200 μl of wash buffer (10 mM Tris-HCl, pH 7.5, 150 mM NaCl, 1 mM MgCl2, 0.5 mM EDTA, pH 8.0, 0.5 mM DTT, 20 ng/μl BSA), and blocked with 500 μl of blocking buffer (500 μl wash buffer + 5 μl of 100 mg/ml BSA + 10 μl of 10 mg/ml yeast tRNA) for 10 min at room temperature with rotation. Then 3 μg of the mouse anti-FLAG M2 antibody (Sigma F1804) or 2 μg of mouse anti-T7 antibody (Millipore Sigma 69522) were added to the dynabeads in blocking solution, and continue to rotate at room temperature for at least 30 min to let the antibody bind to the beads.

For monomer SELEX, the dynabeads with bound antibody were separated from the supernatant and were rinsed twice with wash buffer. To perform a rinse, add 1 ml of the wash buffer to the tube, and invert the tube a few times. The SELEX samples were then applied to the dynabeads and incubated at room temperature for 20 min, with occasional mixing by pipetting. The beads were rinsed three times with wash buffer.

For dimer SELEX, the FLAG-tagged protein was always pulled down the first. The dynabeads with bound M2 antibody were separated from the supernatant and were rinsed twice with wash buffer. Next, the samples were applied to the dynabeads and incubated at room temperature for 20 min, with occasional mixing by pipetting. The beads were rinsed three times with wash buffer. In all, 100 μl of FLAG elution buffer (500 ng/μl 3×FLAG peptide in wash buffer) was used to competitively elute bound protein-DNA complexes for 10 min at room temperature with occasional pipetting to mix.

The dynabeads with bound anti-T7 antibody were magnetically separated from the supernatant and were rinsed twice with wash buffer. The eluate was also magnetically separated from the beads, and loaded to the anti-T7 antibody conjugated dynabeads for the second pull-down. The samples were incubated at room temperature for 20 min with occasional mixing by pipetting. The beads were then rinsed three times, and all supernatant removed after magnetic separation.

To purify bound DNA, 500 μl of wash buffer was used to resuspend the beads, and 25 μl of 20%SDS and 100 μl of 3 M sodium acetate, pH 5.2 were added. Next, the sample was extracted with phenol:chloroform and then with chloroform. In all, 1 μl of 20 mg/ml glycogen was added to the sample, and the DNA was purified by isopropanol precipitation. The DNA pellet was then dissolved with 25 μl of 10 mM Tris-HCl, pH 8.0. Finally, 5 μl of the DNA was used to measure the DNA concentration using Qubit fluorometer with Qubit dsDNA HS Assay Kit (Thermo Fisher Scientific Q32854).

**Generation of sequencing libraries from purified SELEX DNA.** The sequencing libraries were generated by PCR using Phusion DNA polymerase (NEB M0530S, or Thermo F530L). The 50 μl PCR reaction was assembled by mixing 10 μl of 5xHF buffer, 1 μl of 10 mM dNTP, 5 μl of purified SELEX DNA (for R0, 1:5000 dilution was used), 1 μl of 0.5 μM SELEX-for primer and 0.5 μM SELEX-rev primer, 5 μl of 10 μM NEB universal primer and 10 μM NEB index primer (in NEBNext Multiplex Oligos for Illumina, NEB E7335, E7500), 0.5 μl of Phusion DNA polymerase and 21.5 μl H2O. The eight different SELEX-seq primers were designed to increase complexity by sequencing different libraries at different paces. The following program was used for PCR: 1 cycle of 98 °C for 30 s, 5 cycles of 98 °C for 10 s, 60 °C for 30 s and 72 °C for 15 s, 14 cycles of 98 °C for 10 s and 65 °C for 75 s, 1 cycle of 65 °C for 5 min, and holding at 4 °C. The PCR products were purified using 75 μl (1.5 volume) of AMPure XP beads (Beckman Coulter A63881), and eluted with 15 μl of Qiagen EB buffer. The sequencing libraries were analyzed using Bioanalyzer High Sensitivity DNA Analysis (Agilent 5067-4626) and were quantified using Qubit dsDNA HS Assay Kit (Thermo Fisher Scientific Q32854).

**High-throughput sequencing.** All high-throughput sequencing was performed using Illumina NextSeq 500 sequencer, combined with NextSeq 500/550 High Output Kit v2 (75 Cycles) (Illumina FC-404-2005). The individual libraries were normalized to 4 nM and pooled, and then denatured, neutralized, and diluted according to the Illumina NextSeq System Denature and Dilute Libraries Guide before sequencing. The phiX (Illumina FC-110-3001) control library was always

used as an internal control according to the Illumina NextSeq System Denature and Dilute Libraries Guide.

**Bioinformatics**. Four separate FASTQ files were obtained for each library, each coming from one lane of the sequencing run. The 4 FASTQ files of the same library were first concatenated to generate a single FASTQ file, before any further analysis. Mapping of the reads was performed using the galaxy version of bowtie (Galaxy Version 1.2.0 "Map with Bowtie for Illumina" on usegalaxy.org)[59] against the fly genome build dm3, with the parameter -m 1, which means only uniquely mapped reads were kept, and if a read could be mapped to multiple genome loci, it would be suppressed. The resulting SAM files were then filtered (using the Galaxy Version 1.8 + galaxy1 "Filter SAM or BAM, output SAM or BAM" function on usegalaxy.org) to remove unmapped reads. After mapping and filtering, only uniquely mapped reads were kept. ChIP-seq peak calling was performed using the galaxy version of MACS2 (Galaxy Version 2.2.7.1 + galaxy0 "MACS2 callpeak" on usegalaxy.org)[60], with the following setting: --nomodel –extsize 200, and all other parameters were default. ATAC-seq peak calling was also performed with the galaxy version of MACS2, with the following setting: --nomodel –extsize 200 - -shift −100.

To perform peak analyses and motif searches, the bed files containing called peaks were first filtered to remove peaks in heterochromatic regions and those mapped to chrU and chrUextra. The remaining peaks were assigned to genes and different genomic locations (introns, promoters etc.) using homer V4.10 [61] according to genome build dm3, and the pie graphs were generated using Microsoft Excel. de novo motif searches were also performed with homer V4.10 using peaks located in intergenic and intronic regions, with the following parameters: dm3 -size 80 -len 8 -mis 1 -mask.

Differential binding analyses were performed using DiffBind (V2.12.0)[62], with the following parameters: AnalysisMethod=DBA_EDGER, summits=250, minMembers=2. Two biological replicates of each condition to be compared (in total 4 ChIP-seq experiments) were fed into DiffBind, and each biological replicate consisted of both ChIP and input samples. Differential loci were defined as loci being called as peaks by MACS2 in at least two out of four ChIP experiments (minOverlap=2), and having an FDR < 0.05. Common loci were defined as loci being called as peaks by MACS2 in at least three out of four ChIP experiments (minOverlap=3), and having a P value > 0.1. "and/or" loci (for example "all Scr$_{T1}$ and/or Ubx$_{T3}$ loci" and "all ATAC$_{T1}$ and/or ATAC$_{T3}$ loci" in Supplementary Fig. 4) were defined as all loci being called as peaks by MACS2 in at least two out of four ChIP or ATAC experiments. Loci in heterochromatic regions, as well as loci mapped to chrU and chrUextra were removed before performing further analyses.

Differentially accessible loci were obtained in two steps. First, DiffBind was used to find putative differentially accessible loci with the following parameters: AnalysisMethod=DBA_EDGER, summits=250, minMembers=2, FDR < 0.05. Second, from all loci obtained in step 1, those with |log2(Fold)|>1 are reported as differentially accessible loci in Supplementary Data 1.

To sort all 432 Scr$_{T1}$ > Ubx$_{T3}$ peaks according to their Exd-dependency, we compared Scr$_{T1}$ and Scr(YPWM*)$_{T1}$ ChIP results at these loci using DiffBind. According to the log2[fold] values determined by DiffBind, all loci were first divided into those with Scr$_{T1}$ > Scr(YPWM*)$_{T1}$ occupancy, and those with Scr$_{T1}$ < Scr(YPWM*)$_{T1}$ occupancy. The former class was then sorted by FDR in ascending order, and the latter class was sorted by FDR in descending order. The two FDR-sorted classes were then concatenated to obtain the final peak set, which we interpreted as sorted according to Exd-dependency.

To identify Exd-dependent and -independent peaks among all 432 Scr$_{T1}$ > Ubx$_{T3}$ peaks, we manually correlated these 432 peaks with peaks reported in Fig. 4d. The 432 peaks were sorted according to their Exd-dependency determined by FDR values reported by DiffBind. Peak 141 is the last Scr$_{T1}$ > Ubx$_{T3}$ peak that overlaps with a peak in the Scr$_{T1}$ > Scr(YPWM*)$_{T1}$ class (667 in total) we defined in Fig. 4d. Similarly, peak 261 is the first ScrT1 > UbxT3 peak that overlaps with a peak in the Scr$_{T1}$ ≈ Scr(YPWM*)$_{T1}$ class (3338 in total). Therefore, peaks 1–141 were defined as Exd-dependent Scr$_{T1}$ > Ubx$_{T3}$ peaks, and 261–432 as Exd-independent Scr$_{T1}$ > Ubx$_{T3}$ peaks.

**NRLB analysis**. The dm3 genome was analyzed for Scr and Ubx monomer motifs and Scr-Exd and Ubx-Exd dimer motifs using the PSAMs from the R package NRLBtools (V1.1)[18]. Custom code was used to compute relative affinities at every offset in the genome with these models, with the affinities summed across both strands at every position and normalized to the genomic maximum. Normalized relative affinities less than $10^{-4}$ were ignored for further analysis. To smooth out significant position-specific local variations, the remaining affinities were used to construct a windowed maximum across the genome: for every window of length k in the genome, the maximum affinity found within that window was used. Here, k is the length of the PSAM used. These windowed affinities were then stored as bigwig files, effectively converting them to "affinity tracks" to facilitate further downstream analyses. UbxIa and UbxIVa are two Ubx isoforms, and monomer and Exd- dimer models for both isoforms are available in the NRLBtools package. Both isoforms were used in the analyses and gave similar global patterns, and UbxIa results were shown throughout this study to represent Ubx.

New NRLB models from gel-free SELEX data were constructed on the monomer and dimer data using the multi-mode modeling strategy with growth outlined in ref. [18]. For models fit to the monomer data, two modes were used with

starting $k = 7$ and grown to $k = 13$, while models fit to the dimer data used three modes with starting $k = 16$ grown to $k = 18$. In both cases, shift symmetries of length 1 and flank lengths between 0 and 3 were tested, and dinucleotide parameters were added at the end. As the length of the variable region in the monomer data was 16 bp, an R0k of 6 bp was used, while the dimer data used an R0k of 5 bp as the variable region was 24 bp. Of the various models generated for every dataset, the one with the highest likelihood was selected and one mode was selected and displayed in Fig. 7.

**Data visualization**. Bigwig files from MACS2 peak calling were loaded into the genome browser IGV[63] and visually analyzed. Tracks covering selected regions (for example, the enhancers selected to generate reporters) were taken as screen snapshots.

Heatmaps and histograms were generated using the galaxy version of deepTools3[64] with -binSize=20, and the colors of histogram lines were changed with Adobe Illustrator when necessary. Bed files used in generating the heatmaps were from differential binding analyses described above, and the bigwig files used were from MACS2 peak calling of ChIP-seq and ATAC-seq data, or generated from genome-wide NRLB analysis described above. Scr$_{T1}$ and Ubx$_{T3}$ ChIP datasets were normalized with a scaling factor such that the average Scr$_{T1}$ ChIP signal and average Ubx$_{T3}$ ChIP signal were the same at the peak center for Scr$_{T1}$ ≈ Ubx$_{T3}$ loci. A similar scaling factor was computed to normalize Scr(WT) and Scr(YPWM*) ChIP datasets, as well as Dll$_{T1}$ and Dll$_{T3}$ ChIP datasets. No scaling factors were applied to NRLB scores, and the scores reflect normalized relative affinity of each binding mode.

Pearson's Correlation Coefficient (PCC) was computed using the R function cor.test. All scatter plots and density plots were generated in R in two steps. First, for each locus of interest, the scores for ChIP-seq signals, ATAC-seq signals, or NRLB signals were extracted from corresponding bigwig files. To extract a score of a locus, the signal score of each base pair of a 100 bp interval flanking the peak center (from 50 bp upstream to 50 bp downstream) was extracted using R functions, and the sum of the 100 base pair scores was defined as the score of the locus. For ChIP-seq scores, the same scaling factors used for heatmaps were also applied to normalize different ChIP datasets. Second, the plots were generated using the R package ggplot2.

For the plots showing Scr-Exd and Ubx-Exd scores across wild-type and mutant enhancers, each sequence was first analyzed using the R package NRLBtools[18], and a table containing the position and affinity information of ten strongest motifs of each dimer was then generated. Again, the UbxIa-Exd score was used to represent Ubx-Exd affinity, and all relative affinity scores were normalized to the genome max. The final plots were then generated from the tables using the R package ggplot2.

**RNA-seq and data processing**. Imaginal discs were dissected from isogenic $w^{1118}$ (Bloomington #5905) wandering larvae. For each biological replicate, 40 leg discs were dissected. Larvae of mixed sexes were used. The selected larvae were dissected in PBS + 1%BSA (filtered), and the samples were kept cold for as much as possible during dissection. The dissected discs were transferred with a P-20 pipette to a 1.5-ml tube containing 350 μl of the RLT buffer (in RNeasy mini kit, Qiagen 74104) with 1% β-mercaptoethanol, and the discs were immediately homogenized with a plastic pestle. The homogenized samples might be kept at −20 °C for a few days. PBS was added to each sample to make the total volume 450 μl, and 250 μl of pure ethanol was added. The entire samples were mixed well and loaded on RNeasy mini columns. After washing once with 700 μl of RW1, and twice with 500 μl of RPE, the RNA was eluted twice with 45 μl of nuclease-free water, resulting in 90 μl of the eluate.

The RNA was then treated with DNase I (NEB M0303S) to remove trace amount of genomic DNA. 10 μl of the 10× DNase I buffer was added to each sample, and the samples were mixed. 2 μl of DNase I was then added to each sample. The samples were mixed well and incubated in 37 °C water bath for 30 min. The RNA was then cleaned up using the RNeasy micro kit (Qiagen 74004). To each sample, 350 μl of the RLT buffer with 1% β-mercaptoethanol and 250 μl of pure ethanol were added. The samples were mixed well and loaded on RNeasy MinElute Spin Columns. After washing once with 700 μl of RW1, once with 500 μl of RPE, and once with 500 μl of 80% ethanol according to the manufacturer's standard protocol, the RNA was eluted with 15 μl of nuclease-free water. The RNA samples were quantified with nanodrop, and appropriate dilutions were run on bioanalyzer with Agilent RNA 6000 Pico Kit (Agilent 5067-1513) to ensure the RNA samples had high integrity.

RNA-seq libraries were prepared using the NEBNext Ultra II Directional RNA Library Prep Kit for Illumina (NEB E7760S), following the manufacturer's instructions, with the following custom parameters. In all, 50 ng of total RNA was used as starting materials, and 14 cycles of PCR amplification were performed. AMPure XP beads (Beckman Coulter A63881) were used in size selection, and the target mean library size was 520 bp (mean insert size about 400 bp). In total, 22 μl of beads were added in the first bead selection, and 10 μl was added in the second bead selection. In the last step, the library DNA was eluted with 17 μl of 0.1×TE buffer, instead of 23 μl stated in the kit's manual, and 15 μl was transferred to 1.5-ml tubes. The libraries were first quantified with nanodrop, and according to the nanodrop readings, appropriate dilutions were made for Qubit quantification and bioanalyzer analysis. The dilutions were accurately quantified using Qubit

fluorometer with Qubit dsDNA HS Assay Kit (Thermo Fisher Scientific Q32854), and the library sizes were determined by bioanalyzer, using Bioanalyzer High Sensitivity DNA Analysis (Agilent 5067-4626). Finally, the libraries were adjusted to 4 nM each and pooled for sequencing with the Illumina NextSeq 550 sequencer.

The fastq files were processed using tools on usegalaxy.eu. The Cutadapt tool (Galaxy Version 4.0 + galaxy0) was used to trim adaptor sequences from the reads (-a AGATCGGAAGAGCACACGTCTGAACTCCAGTCAC -A AGATCGGAA GAGCGTCGTGTAGGGAAAGAGTGTAGATCTCGGTGGTCGCCGTATCA TT), and to filter out reads shorter than 100 bp (-minimum-length 100). The filtered reads were then mapped to the genome build dm6 using the RNA STAR tool (Galaxy Version 2.7.8a + galaxy0) with the following parameters: --sjdbOverhang 149. The featureCounts tool (Galaxy Version 2.0.1 + galaxy2) was used to obtain the transcript count tables, and the R package DESeq2 (V1.24.0) was used to identify differentially expressed genes.

**Co-immunoprecipitation (co-IP)**. The protein mixture was set up by mixing 20 µl of dialysis buffer with 0.05% Tween-20, 2.5 µl of 1 µM T7-Dll, and 2.5 µl of 1 µM EGFP-Scr-FLAG. In all, 10 µl of 5x conditioning buffer (50 mM Tris-HCl pH 7.5, 250 mM NaCl, 5 mM MgCl2, 2.5 mM DTT, 2.5 mM EDTA, 100 ng/µl BSA, 0.025% Tween-20) was mixed with 15 µl of H$_2$O, and the entire 25 µl buffer was added to the protein mixture to set up the co-IP binding. The binding was performed at room temperature for 30 min.

Overall, 30 µl of Dynabeads™ Protein G (Thermo Fisher 10004D) per binding reaction was rinsed once with wash buffer (10 mM Tris-HCl pH 7.5, 150 mM NaCl, 1 mM MgCl$_2$, 0.5 mM EDTA, pH 8.0, 0.5 mM DTT, 20 ng/µl BSA, 0.05% Tween-20), and blocked with blocking buffer (1 ml wash buffer + 10 µl of 100 mg/ml BSA + 20 µl of 10 mg/ml yeast tRNA) for 10 min at room temperature with rotation. In total, 3 µg of mouse anti-FLAG M2 antibody (Sigma F1804) per co-IP reaction was added to the beads, and continue to rotate at room temperature for at least 30 min to let the antibody bind to the beads.

The dynabeads with bound antibodies were separated from the supernatant and were rinsed twice with wash buffer. The co-IP samples were then applied to the dynabeads and incubated at room temperature for 20 min, with occasional mixing by pipetting. The beads were then washed three times with wash buffer. During the last wash, the samples were transferred to a new tube and the supernatant was removed. In all, 100 µl of 4× SDS-PAGE sample buffer (with 10% β-mercaptoethanol) was used to resuspend the beads. The samples were heated at 95 °C for 5 min, and the supernatant was loaded on SDS-PAGE. The proteins were transferred to the PVDF membrane using routine protocol, and the membrane was blotted with 1:5000 HRP-conjugated anti-T7 antibody (Millipore Sigma 69048). SuperSignal™ West Femto Maximum Sensitivity Substrate (Thermo Fisher 34095) was used to visualize the target protein.

**In vitro pull-down using biotin-labeled DNA probes**. The biotin-labeled DNA probes were generated by annealing a primer with 5' biotin label to the probes that have the primer binding site at their 3' end, followed by Klenow mediated primer extension. In all, 4 µl of 10× STE buffer (see above), 30 µl H$_2$O, 4 µl of 10 µM biotin-SR1 primer, and 2 µl of 10 µM probe were mixed. The annealing was performed using the following thermocycler program: 98 °C for 3 min, 93 cycles of 97 °C for 30 s, with −1 °C per cycle, holding at 4 °C. 10 µl of 10x NEBuffer 2, 8 µl of 10 mM dNTP, 41 µl H$_2$O and 1 µl of Klenow fragment (NEB M0210L) were added to the annealed DNA, and the sample was incubated at 37 °C for 15 min. Overall, 5 µl of 0.5 M EDTA, pH 8.0 was used to stop the reaction. The probes had a concentration of ~200 nM, and were directly used in the pull-down assays without further purification.

The binding reactions were assembled from protein mixtures and probe mixtures. The protein mixture contained 0.1 pmol of Hox protein, and between 0.1 and 0.5 pmol of its binding partner in 500 µl of dialysis buffer with 0.05% Tween-20. In all, 50 µl of 200 nM biotin-labeled probe was added to 200 µl of 5× conditioning buffer (50 mM Tris-HCl pH 7.5, 250 mM NaCl, 5 mM MgCl2, 2.5 mM DTT, 2.5 mM EDTA, 100 ng/µl BSA, 0.025% Tween-20) and 250 µl of H$_2$O to generate 500 µl of the probe mixture. The protein mixture and the probe mixture were combined and incubated at room temperature for 30 min.

In all, 10 µl of Dynabeads™ MyOne™ Streptavidin T1 (Thermo Fisher 65601) was used for each pull-down reaction. The beads were rinsed once with wash buffer (10 mM Tris-HCl, pH 7.5, 150 mM NaCl, 1 mM MgCl$_2$, 0.5 mM EDTA, pH 8.0, 0.5 mM DTT, 20 ng/µl BSA, 0.05% Tween-20), and blocked with blocking buffer (1 ml wash buffer + 10 µl of 100 mg/ml BSA + 20 µl of 10 mg/ml yeast tRNA) for 10 min at room temperature with rotation. The binding reaction was then loaded to the blocked beads and the sample was rotated at room temperature for 20 min. After removing the supernatant, the beads were resuspended with 1 ml of wash buffer, and the entire sample was transferred to a new tube. The supernatant was removed, and the beads were resuspended with 50 µl of 4× SDS-PAGE sample buffer (with 10% β-mercaptoethanol). After heating the sample at 95 °C for 5 min, the supernatant was loaded on SDS-PAGE. Transfer to PVDF membrane was done with standard protocol. 1:1333 mouse anti-FLAG M2 antibody (Sigma F1804) was used as the primary antibody, and 1:1333 HRP-conjugated Goat anti-mouse IgG (Jackson ImmunoResearch 115-035-003) was used as the secondary antibody. SuperSignal™ West Femto Maximum Sensitivity Substrate (Thermo Fisher 34095) was used to visualize the target protein.

**Reporting summary**. Further information on research design is available in the Nature Research Reporting Summary linked to this article.

## Data availability
The data that support this study are available from the corresponding author upon reasonable request. All next-generation sequencing data generated in this study have been deposited in the GEO database under accession code GSE184454. Source data are provided with this paper.

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

## Acknowledgements

We thank K. Monahan for help with ChIP-seq data analysis; J.F. Kribelbauer and C.E. Howard for assistance on data analysis and visualization; R. Delker for help on confocal imaging and image processing; S. Davis for assistance in stereoscope imaging; M. Mannervik for the CBP antibody; and all present and past members of the Mann lab for discussions and comments. This work was supported by NIH grants R35 GM118336 to R.S.M., R01 HG003008 and R01 MH106842 to H.J.B., and F32 GM125329 to W.J.G.

## Author contributions

R.S.M. conceived the study. S.F. and R.S.M. designed the study. R.E.L. generated the ATAC-seq data. S.F. performed all other experiments. C.R. performed NRLB motif analysis with input from S.F. and H.T.R. W.J.G. performed phylogenetic analysis of the dsx−1 CRM and analyzed SELEX-seq results with S.F. S.F. and R.S.M. performed all other data analyses. H.J.B. supervised *NRLB* analysis and provided input and expertise. S.F. and R.S.M. wrote the manuscript. R.S.M. supervised the entire study.

## Competing interests

The authors declare no competing interests.
