## [Peer Review File · Nature Communications]

Reviewers' comments:

Reviewer #1 (Remarks to the Author):

This manuscript by Feng et al. reports on the identification of Dll as a new Hox cofactor in *Drosophila* that enhances Scr binding specificity as compared to Ubx, resulting in Hox binding specificity in a tissue-specific manner. They used their series of data to propose that HOX binding specificity is achieved through their interaction with cofactors but with distinct modus operandi depending on the cellular context. While this study is of potential interest for researchers interested in Hox biology, there are several issues with this manuscript that damp my enthusiasm for this manuscript.

General comments:

1. I was really surprised that the authors do not discuss the evidence that Dll suppress Exd function in this organ (Gonzales-Crespo et al., Nature 1998). I feel that this is an important finding and the authors should integrate it in the interpretation of their data.
2. As far as I know, the ground state of the leg imaginal disc is the antenna imaginal disc and I don't really understand on which basis the authors state that T2 leg is the ground state.
3. In the same line of idea, the authors should take into account in the interpretation of their data that Antp is repressed by Dll and reduced expression of Antp is required for proper development of the distal leg (Emerald and Cohen, 2004).
4. It would be useful to have a co-staining for Scr and Exd and importantly the difference in expression pattern for Scr (uniform) and Exd (restricted) should be taken into account in the molecular analyses that are based on the use of the entire imaginal disc.
5. To support the statement made by the authors that Dll is a Scr co-factor, it would be important to show that Dll indeed physically interact with Scr or, at least, that Dll and Scr have common targets in the leg imaginal disc.
6. The authors propose that Scr and Dll could act as repressor at their target CRM. However they do not provide any experimental evidence to support this statement.
7. Regarding the FLAG ChIP-seq experiments, the authors use FLAG ChIP from wild type tissue as control. This is not an appropriate control. It would be important to use imaginal discs (or at least a cell line) in which FLAG is ubiquitously expressed.
8. No scale is provided for the ChIP-seq data and as ChIP-seq is not a quantitative assay the authors should refrain from qualifying peaks as being 'stronger' or 'weaker' peaks.
9. There is no Exd ChIP to address binding sites as Exd dependent and independent sites. YPWM motif is important for many more protein-protein interactions and not just for Scr-Exd.

Reviewer #2 (Remarks to the Author):

In this manuscript, the authors explore how two different Hox proteins – Scr and Ubx – function within serially homologous structures – the first (T1) versus the third (T3) thoracic leg imaginal discs – to bind DNA and regulate gene expression. This is a rather detailed analysis, so bear with me in the description of their findings.

Scr is normally expressed in and required to make the T1 leg disc different from the second thoracic (T2) leg disc (considered the "ground state" for leg discs). Ubx is normally expressed in and required to make the T3 leg disc different from the T2 leg disc. Sequencing of the transcripts from the 1st and 3rd leg discs reveal that only a few genes show greater than two-fold differences in expression, and those few genes include both Scr and Ubx. The authors don't do anything further with this data (as far as I can tell) except argue that the differences between the leg primordia must largely be due to the timing/levels of target gene expression.

The authors then carry out ChIP-seq using 3X FLAG tagged versions of each protein (tagged at the

endogenous locus to look at what each protein binds to in the respective leg discs. They find that the two proteins bind to many shared loci (5346) well as to more limited leg disc specific loci: 432 loci are bound to higher levels by Scr in T1 (ScrT1 >UbxT3) and 62 loci are bound to higher levels by Ubx in T3 (UbxT3>ScrT1). They show that both ScrT1 >UbxT3 sites and UbxT3>ScrT1 sites map more frequently to intergenic/intron regions than do the ScrT1=UbxT3.

The authors do ATAC of T1 and T3 legs and find very little difference in gene/chromatin accessibility whether the gene is bound the same by both proteins in both leg discs or shows preferential binding in each specific leg disc, but there are small handful of genes for which accessibility seems to change. Interestingly, it is in the Antp-complex (where Scr is found) for ScrT1 >UbxT3 and in the Bx-C (where Ubx is found) for UbxT3>ScrT1. They conclude that chromatin accessibility does not account for the differential Hox binding in the two discs.

Using a computational method called "No read left behind (NRLB)", which transforms SELEX in vitro DNA binding data into binding specificities for a given TF, they determine binding affinities for the Hox monomer and Hox:Exd heterodimers and then plot the average relative affinity to binding peaks comparing ScrT1 >UbxT3 to ScrT1 =UbxT3 for Scr. They state that monomer binding sites are not much more enriched in Scr preferred sites over shared sites, but, they state (*see comments section) that there are significant enrichments of Scr-Exd dimer motifs centered on binding events where ScrT1 >UbxT3 compared to binding events where ScrT1 =UbxT3, suggesting that Scr-Exd heterodimers might contribute to the differential binding of ScrT1.

They test this idea by comparing WT Scr ChIP-Seq to ChIP-Seq with an Scr mutant in which the Exd binding motif (YPWM) has been deleted. Binding does not change for most sites (dubbed Exd-independent sites), but decreases in some (667 loci they call Exd-dependent) and increases in others (360 loci), which they suggest may reflect increases in Scr monomer binding. Using the NRLB binding affinity program, they show that the reduced binding correlates with sites that have a higher Scr-Exd binding affinity (*again see comments) and the increased binding correlates with sites that have increased Scr monomer binding affinity. Indeed, de novo motif discovery identified Hox-Exd motifs with WT bound DNA, but with the YPWM mutant the top site was an Scr monomer binding motif. 42% of loci with ScrT1 > UbxT3 binding were Exd-dependent (binding > 2X with mutant), whereas only 28% of loci with equivalent ScrT1 = UbxT3 binding were Exd-dependent. 32% of ScrT1 > UbxT3 were Exd independent, whereas 65% of ScrT1 = UbxT3 were Exd dependent. Altogether, this suggests that Hox-Exd dimer binding accounts for some of the differences in binding of Hox proteins in the two tissues.

They then built 25 reporter constructs for ScrT1 > UbxT3 loci that contained the Scr-Exd binding sites. Eight of these constructs showed higher reporter gene expression in T1 than in T3. They mutated the binding sites in three of them and lost the differential T1 expression.

Since other factors must explain the other differentially bound ScrT1 > UbxT3 loci , they explored other potential co-regulators, settling on Dll as a good candidate. They did ChIP-seq with Dll and found a nice correlation with Exd-independent Scr binding peaks. They did SELEX to get data for their "no read left behind" program to identify what Dll binds alone and in the presence of Scr (as well as running SELEX on Scr and Scr +Exd). Based on the Scr-Dll motif, they did in vitro DNA binding assays of Scr with and without Dll on their optimal NRLB site and two candidate target sites in the genome. They built reporter constructs again (9 of them), two showed difference in T1 versus T3. They mutated one of the shared Scr/Dll sites in the dsx-1 reporter construct and lost expression in male T1 discs. They mutated one of the shared peaks in the dpy-1 enhancer and lost expression completely.

Comments:

In general, this is a nice study that follows up on several published papers (mostly from the same lab) showing that Hox::Exd heterodimers versus Hox monomers provides the specificity of relevant gene

binding – i.e. the binding that leads to differential gene expression driven by different Hox genes.

The authors have some nice controls in this study, including using the same tag and antibody to do the ChIP-Seq and tagging the genes at the endogenous loci. They also include background binding with a genotype negative control.

There are a few items that need to be addressed for this study to advance the field.

1. They should define CRMs the first time the abbreviation is used.
2. The authors should indicate why binding to intergenic and intronic regions is more relevant than binding to other parts of genes. When the authors show the total binding profiles for both Hox proteins (Figure 1G, 4B), it looks like binding near the TSS predominates – is that binding not meaningful? They should make a clear argument for why binding in certain regions is more relevant to gene regulation (if such arguments exist) and then make it more obvious which binding events are being considered when they are determining binding motifs (i.e. for their de novo discovery of motifs).
3. Somewhere the authors should indicate what they mean by TSS (how many nucleotides from the TSS do binding events have to be to fall into this category?).
4. The change in the scales on the two graphs in Fig2d and Fig2e, and the much lower average relative affinities for the heterodimer sites relative to the monomers suggest (1) an artificial exaggeration in the differences in binding (in fact, in absolute terms the difference in monomer binding in ScrT1 >UbxT3 versus ScrT1 =UbxT3 is greater than the difference in heterodimer binding affinities for the same two groups) and (2) that the low affinity binding sites need to be compensated by relatively higher concentrations of the heterodimers, if they are relevant to gene expression. This should be discussed when the data are first reported on and dealt with in the discussion – see later comment.
5. Figure 4D top graphs do not have scale bars. The image looks like the scale bars are probably the same, but I'm betting that, based on Figure 2d and 2e, they are not.
6. I was initially expecting the Exd-binding (YWPR) mutant of Scr to behave more like a null with respect to the number of sex comb teeth; this is explained later in the paper where we are reminded that Exd is only nuclear (and therefore functional) in the Hth-expressing (peripheral/future proximal) domain of the leg discs. Thus, loss of formation of Exd heterodimers would not be expected to affect sex combs since they form on a more distal leg segment. It would be helpful for the authors to say something about not expecting the sex comb number to be affected since that structure forms outside the Exd nuclear domain. Also, I'm not sure the same is true regarding the suppression of scutellar bristles in T1. Are these Exd-independent structures? Don't they form in the Exd nuclear domain?
7. In the reporter gene constructs, the exact mutations that were made were not shared (as far as I can find in the paper). It seems like the mutations were complete deletions of the Scr-Exd sites. If so, THIS IS NOT THE RIGHT EXPERIMENT. If the point of this paper is that the differential binding of Hox::Exd and not binding of the homodimers that makes a difference in which genes are differential expressed in the two leg discs, then they need to replace the Scr-Exd site with an Scr monomer site and ask happens to reporter gene expression.
8. Likewise, if they are building a case that it's the partner proteins that make for specificity – they need to replace the Scr-Dll site with an Scr monomer site instead of deleting the site entirely. Here, again, it is not obvious what got deleted.
9. I would like some reconciliation between this study and what has been discovered in the past. Is there any example out there showing that Hox monomers can regulate gene expression on their own?

I think it has been nicely established that Hox::Exd heteromers can regulate gene expression. If the binding affinity for Hox::Exd sites is really about 8X lower than that for Hox monomers, how are the heterodimer binding sites relevant? How do Hox proteins behave in the presence of Exd (with or without DNA) – do they preferentially form dimers?

10. It would also be helpful to correlate the binding data to the RNAseq data. Are the ScrT1=UbxT3 bound genes expressed to equal levels in the T1 versus T3 disc. Are the ScrT1>UbxT3 expressed to higher levels in the T1 disc? Do the sites that are Exd-dependent correlate better with gene expression? How about the Scr-Dll sites? They have the RNAseq data and they have the binding data; seems like a straightforward analysis. I realize that binding does not equal gene expression, but in the end, it's the differences in gene expression that matter and only relating binding to regulation for a few chosen reporter genes (and then inadequately addressing the big story in the paper of monomer versus heterodimer binding) seems inadequate given the data they have at hand.

Reviewer #3 (Remarks to the Author):

In the current study, the authors try to address a significant developmental biological question- How do Hox proteins which have similar DNA binding properties in-vitro initiate paralog-specific gene regulatory networks in-vivo? Using the T1 and T3 legs in *Drosophila* as models which are specified by the Hox proteins Scr and Ubx respectively, the authors try to understand the mechanisms which govern the differences between the two structures. They perform ChIP-sequencing for tagged Scr and Ubx proteins in T1 and T3 leg imaginal discs and identify close to 8% of binding loci which are paralog specific. Using ATAC seq experiments and a bioinformatics tool (NRLB), the authors arrive at the conclusion that paralog-specific binding events are a result of Scr-Exd interactions. However, ChIP-sequencing for a mutant version of Scr protein which cannot bind Exd indicate that only a handful of Scr bound loci dependent on Exd. The authors, however, further pursue the idea that Exd is significantly responsible for paralog specific binding of Scr in T1 and justify this idea using other bioinformatics, mechanistic and transgenic assays. Further, the authors identify Dll as a novel cofactor of Scr which helps in Scr binding in regions where Exd is not expressed. Taken together, the authors claim to identify paralog-specific binding loci for the Scr protein in the T1 leg discs and try to justify that binding with cofactors like Exd and Dll are critical for such paralog-specific binding.

Conceptual Issues

1. RNA sequencing analysis for the T1 and T3 leg imaginal discs suggest that only a handful of genes are differentially expressed between the two discs, notable of which were Scr and Ubx themselves. Previous studies have suggested that Ubx regulates the expression of Scr in T3 legs. This could suggest two possibilities: 1) differences between the T1 and T3 legs might occur due to subtle changes in the gene regulatory network which might manifest from differences downstream of Hox targets and not at the level of Hox targets themselves. 2) differences between T1 and T3 legs might be due to events occurring during the pupal stage. The T1 and T3 leg imaginal discs, thus, do not seem to constitute a sound system to establish the causal relationship between paralog specific binding of Hox proteins to the differences in phenotype observed between the T1 and T3 legs.

Analytical and validity issues and suggested improvements

Figure 1 and 2:

The authors perform ChIP-sequencing analysis for Ubx and Scr in T3 and T1 leg imaginal discs respectively. They identify close to 8% of binding loci which is bound differentially by Scr but not by Ubx. Additionally, the paralog specific binding sites observed by the authors is largely at intergenic regions.

There are a few issues with the results and interpretations here:

1. The term binding events, sites, loci and targets seem to be used very casually and are not consistent. This needs to be taken care of.

2. Even though the authors do not provide a list of the targets that correspond to the differentially bound loci, they claim that there is a strong asymmetry between paralog specific targets of Scr and Ubx in T1 and T3. Additionally, it needs to be investigated whether such paralog-specific binding events result in any differential gene expression between the T1 and T3 discs. Thus, claiming that differences in binding of the Hox proteins translates to gene regulatory changes seems premature.

3. The authors indicate that most paralog-specific binding events occur at the intergenic regions and that CRMs are rich in paralog-specific loci. While the authors do explain in detail the methodology of their assay, a mere 8% paralog specific binding loci (most of which are in intergenic regions) can easily be attributed to non-specific binding, as is usually the case in all genome wide studies. The authors need to be careful here while making such claims and should provide sufficient data to back their statements.

4. De-novo motif analysis of peaks from Scr and Ubx ChIP-seq suggest enrichment of motifs which the authors describe to be Hox-Exd motifs. However, later analysis (see point 7, point 8) indicate that majority of the Scr bound sites are Exd independent. This seems to be contradictory and needs to be properly addressed. The authors should, additionally, make sure that what they claim to be as Hox-Exd sites based on SELEX studies are actually Hox-Exd sites in-vivo also using proper experimental evidence.

5. The fact that binding sites for Ubx in T3 are much lower than Scr in T1 is very interesting. Previous studies have shown that development of the T3 mesothorax is achieved in part by repression of Scr by Ubx. Analyzing whether the paralog specific binding of Ubx in T3 is at the Scr loci can interesting clues to mechanisms governing differences in developmental programs between T1 and T3.

6. In Line 156-158, the authors report that both difference in gene expression as well as chromatin accessibility were found to be different for the Scr and Ubx loci. This would somewhat indicate that regulation of one Hox protein by the other is a major determinant of T1 and T3 leg specification. The minor differences in gene regulatory networks between T1 and T3 can be due to events downstream of Hox direct targets or during the pupariation rather than a direct consequence of paralog-specific binding. To counter this argument, as already suggested, the authors should provide a causal relationship of paralog-specific binding to the difference in the T1 and T3 leg phenotypes.

7. The authors next turn to NRLB model to identify whether monomeric Hox binding motifs or Hox-Exd motifs are enriched in paralog-specific loci. In Line 170-174, the authors observe a similar enrichment score for Scr and Ubx monomeric binding at the ScrT1 \approx UbxT3 loci. Interestingly, they find a similar result for ScrT1>UbxT3 loci. They also find that relative enrichment of Scr-Exd motif is much greater ScrT1>UbxT3 loci as compared to ScrT1 \approx UbxT3 loci, thereby inferring that Exd might have an important role to play in identifying paralog specific targets.

There are two issues with the results here. First, it looks like there is disparity between de-novo search algorithms (which finds Hox-Exd motif to be the enriched motif) and the NRLB models (which finds monomeric Hox motifs to be enriched at the peaks). Second, authors completely ignore the fact that the relative enrichment of monomeric binding sites at ScrT1 \approx UbxT3 loci as well as ScrT1>UbxT3 loci doesn't necessarily mean that they cannot impart specificity of binding. The expression of the Hox proteins and their relative dosage are other factors which can influence Hox specificity of binding to these loci.

Thus, the statement that Exd significantly contributes to paralog-specific Hox binding, at best, seems premature. The authors do not show substantial validations to support such a claim.

Fig 4 and Fig 5:

8. The authors next move on to generate a Scr mutant (Scr (YPWM*)T1) that is unable to interact with Exd to test if heterodimerization with Exd contributes to paralog specific binding. Interestingly, the authors here compare the ChIP-seq for the wild type and mutant Scr and report that majority of sites are Exd independent. De-novo motif analysis for peaks in Scr (YPWM*)T1 seems to indicate that monomeric Scr binding site is the preferred motif.

There are a few inconsistencies and issues with the finding here. First, the very fact that de-novo motif analysis for the wild type Scr shows enrichment of what is termed as the Scr-Exd motif, even when most sites are Exd independent, seems very contradictory. Second, while the authors try to support their finding saying that in the Scr (YPWM*)T1 mutant, the Scr-Exd motif is not enriched, the data in Extended figure 3b is hardly encouraging. The authors seem to ignore the first motif which is very similar to what they term as the Scr-Exd motif and is highly enriched. While the authors should take negative results into account and arrive at their conclusions based on what is observed, I feel it is time to redefine what is suggested as Hox-Exd motif and what as Hox monomeric motif based on experimental data like ChIP-seq and not simply based on in-vitro Selex seq studies. The authors should not make flamboyant conclusions (like the one made in Line 226-228) while not properly taking their results into consideration as well as not providing enough experimental evidence to back their claims.

9. The authors next move onto confirm the importance of Scr-Exd interactions on paralog specific binding events. In line number 237-238, the authors suggest that of the 432 ScrT1>UbxT3 loci, 183 sites seem to be Exd dependent as suggested from ScrT1/Scr(YPWM*)T1 ChIP-signal ratio. Later they do the same analysis for ScrT1≈UbxT3 peaks.

This is where it gets really confusing as the authors suggest that 1545 of 5346 peaks have ScrT1/Scr(YPWM*)T1 ChIP-signal ratio greater than 2. However, previous comparison between ChIP-seq for ScrT1 and Scr(YPWM*)T1 suggested that only 667 binding sites were found to be Exd dependent. This data does not reconcile.

Also, in Figure 5b, the heatmaps are plotted for 2152 peaks? There is no mention of where they got this number from. The authors should make sure that their data is consistent in the text as well as on the images.

Figure 6

10. The authors then go on to generate lacZ reporters to reconfirm the paralog specific activity of Scr and its dependence on Exd using a transgenic approach. While their results do indicate that a fraction of Scr paralog specific loci drive T1>T3 expression, the authors do not provide substantial evidence to claim that such differences are due to the direct regulation by Scr-Exd complex. To identify whether Exd has a role here, the authors should perform genetic assays (perhaps by making Exd clones or showing Exd occupancy using ChIP-seq) rather than just mutating the Scr-Exd sites, as it is somewhat evident from the rest of the data that monomeric Scr might also have a significant role to play in paralog-specific target selection via motifs that, although are referred to as Scr-Exd sites, might as well be high affinity binding motifs for the Scr protein itself.

Overview:

We thank the referees for their constructive comments, which have greatly improved the manuscript. They noticed some inconsistencies and confusing wording, which we have now eliminated. Before getting too detailed, we would like to emphasize the two main significant findings presented here, which merit its publication in *Nature Communications*: **First, we use genome-wide approaches to rigorously test the Hox cofactor and latent specificity hypotheses.** The tests of these hypotheses included here are much more stringent than most of the previous work on Exd/Pbx, which typically relied on reporter genes to study just a few targets. **Second, we show for the first time that another homeodomain protein, Distalless (DII), also functions as a Hox cofactor.** Although it has long been recognized that Exd/Pbx cannot be the only Hox cofactors, no one had come up with alternatives. Thus, our findings fill a large gap in our understanding of how Hox proteins achieve specificity in vivo. More generally, they suggest that there may be many more cofactors used by Hox proteins in vivo.

We took the reviewers' comments seriously, which is why this revision has taken significant time to prepare. As a consequence, we have not only modified the text but have included additional data and analyses to strengthen our conclusions. Therefore, we believe this version effectively addresses all of their concerns.

Here is a summary of new data and/or analyses that were added to the revised manuscript.

1. We performed ChIP against Creb Binding Protein (CBP) (Fig 5b), a known marker for CRM activity, in T1 and T3 leg discs. The CBP binding pattern suggests that Scr>Ubx occupied CRMs are active enhancers and silencers. These data directly address the reviewers' request that we provide evidence that the paralog-specific binding we measure is functional.
2. For additional evidence that paralog-specific binding leads to functional differences in gene expression, we systematically compared our RNA-seq and ChIP-seq datasets (Extended Data Fig 5). This analysis shows that Scr>Ubx binding is associated with genes that often show T1>T3 expression. Further, this analysis suggests that Exd-dependent Scr target CRMs tend to be enhancers, as opposed to silencers, which agrees with previous studies indicating that Hth/Exd are most typically used for gene activation.
3. We performed ChIP against Hth (a surrogate for Exd because Exd is only nuclear when Hth is present) in T1 leg discs (Fig 4d and 5a). The Hth occupancy pattern provides additional evidence that a subset of Scr targets require Exd/Hth for binding. In particular, we find that Hth is bound most strongly to the same sites that also depend on Scr's YPWM motif.

4. We added several antibody stains of leg discs, e.g. Hth staining (Fig 3f and Fig 5c), that the reviewers requested, to show its relationship to Scr and Dll expression and to some of the Scr target genes that we characterize.

5. We added data to show that Scr and Dll can physically interact with each other.

Below we provide a point-by-point response to all reviewers' comments and questions.

Reviewer #1 (Remarks to the Author):

This manuscript by Feng et al. reports on the identification of Dll as a new Hox cofactor in *Drosophila* that enhances Scr binding specificity as compared to Ubx, resulting in Hox binding specificity in a tissue-specific manner. They used their series of data to propose that HOX binding specificity is achieved through their interaction with cofactors but with distinct modus operandi depending on the cellular context. While this study is of potential interest for researchers interested in Hox biology, there are several issues with this manuscript that damp my enthusiasm for this manuscript.

General comments:

1. I was really surprised that the authors do not discuss the evidence that Dll suppress Exd function in this organ (Gonzales-Crespo et al., Nature 1998). I feel that this is an important finding and the authors should integrate it in the interpretation of their data.

This is an interesting question, but not particularly relevant to the paper, which focuses on how Exd and Dll are used as Hox cofactors in distinct parts of the leg imaginal disc after their domains are established. Nevertheless, we are happy to mention how these two domains develop over time and have modified the text accordingly (line 332).

2. As far as I know, the ground state of the leg imaginal disc is the antenna imaginal disc and I don't really understand on which basis the authors state that T2 leg is the ground state.

Because the term 'ground state' has a complicated history, we avoid using the term in the revised manuscript (line 108).

That said, here is some clarification for the reviewer. Struhl (PNAS, 1982) showed that when Ubx function is removed from T3 or when Scr function is removed from T1, the leg identities shift to a T2 leg fate. This has been argued by some to mean that Ubx and Scr work on a T2 ground state to diversify leg identity. But Struhl also showed that removing Antp from T2 (or Scr+Antp from T1 or Ubx+Antp from T3) resulted in a leg to antenna transformation. This observation led to the idea that the antenna is the ground state. However, later work from our lab (Casares and Mann, Science 2001) showed that when the antenna selector gene *homothorax* (*hth*) is also removed from the leg (so, for example, *hth*+Antp from the T2 leg), the appendage identity reverts back to a rudimentary leg fate (same for removing *hth* from the

antenna). We therefore consider a leg-like structure to be the true ground state and that *hth* transforms that identity to antenna and the thoracic Hox genes *Scr*, *Antp*, and *Ubx* take that antennal identity to the three different leg fates.

Although these are interesting points, they again are not especially relevant to the main points in the paper.

3. In the same line of idea, the authors should take into account in the interpretation of their data that *Antp* is repressed by *Dll* and reduced expression of *Antp* is required for proper development of the distal leg (Emerald and Cohen, 2004).

Again, although interesting, that *Dll* is a repressor of *Antp* is not relevant to the idea that *Dll* is an *Scr* cofactor in the T1 leg. Our concern is that describing all of this will be confusing to the reader, so we prefer to keep the paper focused on *Dll* as a Hox cofactor.

4. It would be useful to have a co-staining for *Scr* and *Exd* and importantly the difference in expression pattern for *Scr* (uniform) and *Exd* (restricted) should be taken into account in the molecular analyses that are based on the use of the entire imaginal disc.

Agreed; we have now included such co-stains (Fig 5c).

5. To support the statement made by the authors that *Dll* is a *Scr* co-factor, it would be important to show that *Dll* indeed physically interact with *Scr* or, at least, that *Dll* and *Scr* have common targets in the leg imaginal disc.

Agreed. We have now included protein-protein interaction and co-IP assays for *Scr* and *Dll* (Fig 5e).

6. The authors propose that *Scr* and *Dll* could act as repressor at their target CRM. However they do not provide any experimental evidence to support this statement.

The reviewer is referring to the *dpy-1* CRM, which is expressed in T3 and not in T1. This observation raises the possibility that *Scr* may repress the activity of this CRM in T1. However, when we deleted the binding site for *Scr+Dll*, expression was eliminated from T3 (and not derepressed in T1), suggesting that the deleted sequences are required for CRM activity in T3 and thus provide no information about its function in T1. We acknowledge the limitations of these findings. In the revised manuscript, we made this clear and pointed out alternative explanations to our observations (line 410).

7. Regarding the FLAG ChIP-seq experiments, the authors use FLAG ChIP from wild type tissue as control. This is not an appropriate control. It would be important to use imaginal discs (or at least a cell line) in which FLAG is ubiquitously expressed.

We respectfully do not agree with this comment. The best negative control for all of the FLAG-tagged Hox ChIPs is a ChIP from discs that have no FLAG epitope, which controls for any potential background that the anti-FLAG antibody (or other reagents) may have.

8. No scale is provided for the ChIP-seq data and as Chip seq is not a quantitative assay the authors should refrain from qualifying peaks as being 'stronger' or 'weaker' peaks.

We believe this comment refers to line 393 of the revised manuscript, where we compared Dll occupancy at the same genomic locus in T1 vs. T3 leg discs. This is the only place in the manuscript that we describe one peak as being stronger, which is based on the number of sequencing reads. It is generally accepted that the TF occupancy at the same locus under different conditions can be compared if the same antibody is used in the ChIP experiments, and if the ChIP results being compared are normalized. The comparison we describe here meets these conditions.

9. There is no Exd Chip to address binding sites as Exd dependent and independent sites. YPWM motif is important for many more protein-protein interactions and not just for Scr-Exd.

The only YPWM interacting protein described in the literature, by us as well as many other labs, is Exd.

Nevertheless, in the revised manuscript, we have included Hth ChIP experiments to visualize Exd occupancy (Fig 4d and 5a). Hth is a surrogate for Exd, because the uniformly expressed Exd is only present in the nucleus when it forms a heterodimer with Hth, whose expression is spatially and temporally regulated. The Hth ChIP results provide independent support for the Exd dependency of a subset of Scr targets. In particular, the data shown in Fig 5a reveal a beautiful correlation between dependency on the Hox YPWM motif and the binding of Hth, which both independently point to the presence of Exd.

Reviewer #2 (Remarks to the Author):

In this manuscript, the authors explore how two different Hox proteins – Scr and Ubx – function within serially homologous structures – the first (T1) versus the third (T3) thoracic leg imaginal discs – to bind DNA and regulate gene expression. This is a rather detailed analysis, so bear with me in the description of their findings.

Scr is normally expressed in and required to make the T1 leg disc different from the second thoracic (T2) leg disc (considered the “ground state” for leg discs). Ubx is normally expressed in and required to make the T3 leg disc different from the T2 leg disc. Sequencing of the transcripts from the 1st and 3rd leg discs reveal that only a few genes show greater than two-fold differences in expression, and those few genes include both Scr and Ubx. The authors don't do anything further with this data (as far as I can tell) except argue that the differences between the leg primordia must largely be due to the timing/levels of target gene expression.

We agree that whole disc transcriptome measurements are limited because they cannot reveal small differences in gene expression. Although there are a few exceptions, differences in leg morphology are likely due to differences in the pattern that target genes are deployed, not that they are regulated in an on/off manner. The point of the transcriptome measurements here was to make this point in a more quantitative manner.

Nevertheless, to see if these data can provide evidence for the function of some of the Scr (and YPWM) dependent binding events, we have carried out a computational analysis to look for correlations between our ChIP-seq and RNA-seq results (Extended Data Fig 5). These results suggest that Scr could act as either a transcription activator or a transcription repressor. Further, we found that YPWM-dependent Scr target CRMs are more likely to be enhancers than silencers, an observation consistent with previous reports that Hth (an Exd surrogate) acts as a transcriptional activator.

The authors then carry out ChIP-seq using 3X FLAG tagged versions of each protein (tagged at the endogenous locus to look at what each protein binds to in the respective leg discs. They find that the two proteins bind to many shared loci (5346) well as to more limited leg disc specific loci: 432 loci are bound to higher levels by Scr in T1 (ScrT1 >UbxT3) and 62 loci are bound to higher levels by Ubx in T3 (UbxT3>ScrT1). They show that both ScrT1 >UbxT3 sites and UbxT3>ScrT1 sites map more frequently to intergenic/intron regions than do the ScrT1=UbxT3.

Correct.

The authors do ATAC of T1 and T3 legs and find very little difference in gene/chromatin accessibility whether the gene is bound the same by both proteins in both leg discs or shows preferential binding in each specific leg disc, but there are small handful of genes for which accessibility seems to change. Interestingly, it is in the Antp-complex (where Scr is found) for ScrT1 >UbxT3 and in the Bx-C (where Ubx is found) for UbxT3>ScrT1. They conclude that chromatin accessibility does not account for the differential Hox binding in the two discs.

Correct.

Using a computational method called “No read left behind (NRLB)”, which transforms SELEX in vitro DNA binding data into binding specificities for a given TF, they determine binding affinities for the Hox monomer and Hox:Exd heterodimers and then plot the average relative affinity to binding peaks comparing ScrT1 >UbxT3 to ScrT1 =UbxT3 for Scr. They state that monomer binding sites are not much more enriched in Scr preferred sites over shared sites, but, they state (*see comments section) that there are significant enrichments of Scr-Exd dimer motifs centered on binding events where ScrT1 >UbxT3 compared to binding events where ScrT1 =UbxT3, suggesting that Scr-Exd heterodimers might contribute to the differential binding of ScrT1.

Correct.

They test this idea by comparing WT Scr ChIP-Seq to ChIP-Seq with an Scr mutant in which the Exd binding motif (YPWM) has been deleted. Binding does not change for most sites (dubbed Exd-independent sites), but decreases in some (667 loci they call Exd-dependent) and increases in others (360 loci), which they suggest may reflect increases in Scr monomer binding. Using the NRLB binding affinity program, they show that the reduced binding correlates with sites that have a higher Scr-Exd binding affinity (*again see comments) and the increased binding correlates with sites that have increased Scr monomer binding affinity. Indeed, de novo motif discovery identified Hox-Exd motifs with WT bound DNA, but with the YPWM mutant the top site was an Scr monomer binding motif. 42% of loci with ScrT1 > UbxT3 binding were Exd-dependent (binding > 2X with mutant), whereas only 28% of loci with equivalent ScrT1 = UbxT3 binding were Exd-dependent. 32% of ScrT1 > UbxT3 were Exd independent, whereas 65% of ScrT1 = UbxT3 were Exd dependent. Altogether, this suggests that Hox-Exd dimer binding accounts for some of the differences in binding of Hox proteins in the two tissues.

Correct!

They then built 25 reporter constructs for ScrT1 > UbxT3 loci that contained the Scr-Exd binding sites. Eight of these constructs showed higher reporter gene expression in T1 than in T3. They mutated the binding sites in three of them and lost the differential T1 expression.

Correct.

Since other factors must explain the other differentially bound ScrT1 > UbxT3 loci, they explored other potential co-regulators, settling on Dll as a good candidate. They did ChIP-seq with Dll and found a nice correlation with Exd-independent Scr binding peaks. They did SELEX to get data for their “no read left behind” program to identify what Dll binds alone and in the presence of Scr (as well as running SELEX on Scr and Scr +Exd). Based on the Scr-Dll motif, they did in vitro DNA binding assays of Scr with and without Dll on their optimal NRLB site and two candidate target sites in the genome. They built reporter constructs again (9 of them), two showed difference in T1 versus T3. They mutated one of the shared Scr/Dll sites in the dsx-1 reporter construct and lost expression in male T1 discs. They mutated one of the shared peaks in the dpy-1 enhancer and lost expression completely.

Comments:

In general, this is a nice study that follows up on several published papers (mostly from the same lab) showing that Hox::Exd heterodimers versus Hox monomers provides the specificity of relevant gene binding – i.e. the binding that leads to differential gene expression driven by different Hox genes.

The authors have some nice controls in this study, including using the same tag and antibody to do the ChIP-Seq and tagging the genes at the endogenous loci. They also include background binding with a genotype negative control.

Thank you for an accurate summary of the paper and for the positive comments.

There are a few items that need to be addressed for this study to advance the field.

1. They should define CRMs the first time the abbreviation is used.

Agreed, the text has been modified accordingly (line 136).

2. The authors should indicate why binding to intergenic and intronic regions is more relevant than binding to other parts of genes. When the authors show the total binding profiles for both Hox proteins (Figure 1G, 4B), it looks like binding near the TSS predominates – is that binding not meaningful? They should make a clear argument for why binding in certain regions is more relevant to gene regulation (if such arguments exist) and then make it more obvious which binding events are being considered when they are determining binding motifs (i.e. for their de novo discovery of motifs).

We follow a commonly accepted model that transcription factors (including Hox) function mainly by binding to regulatory regions that are typically separate from promoters. Supporting this model, various studies have shown that CRMs are frequently located in intergenic and intronic regions (for example PMID: 27575958 and PMID: 24896182). In a systematic study of >3000 active CRMs (PMID: 24896182), the authors showed that in *Drosophila*, the median distance between a functionally verified CRM and the promoter it regulates is 10 kb, and only about a quarter of CRMs regulate promoters less than 4 kb away. These results indicate that most CRMs are not close to promoters. We have cited these two references in the revised manuscript.

Indeed, a significant portion of Hox binding is found near promoter regions (TSS regions). We do not think Hox binding at promoter regions is without meaning. A subset of the Hox ChIP peaks in annotated promoter/TSS regions could be from binding to promoter proximal CRMs, while others might be due to DNA loops that bring together Hox bound CRMs and the promoters they regulate. There might also be “non-specific” peaks due to the highly accessible and AT rich (Hox monomer motifs are AT rich) nature of the promoters.

For these reasons, de novo motif searches were performed on intergenic and intronic binding sites only, as stated in the Methods. We excluded other regions (mainly TSS/promoter) because we wanted the search to be focused mainly on regulatory regions, not sequences that may be indirectly bound to CRMs or have other, promoter-specific functions. This information is also in the legend of Fig. 1h.

3. Somewhere the authors should indicate what they mean by TSS (how many nucleotides from the TSS do binding events have to be to fall into this category?).

In our manuscript, the TSS region is defined as -1 kb to +100 bp from the +1 nucleotide of mRNA (the exact transcription start site). This is the default setting of *Homer*, which we used in this study to assign ChIP peaks. This has been specified in the legend of Fig 1g of the revised manuscript.

4. The change in the scales on the two graphs in Fig2d and Fig2e, and the much lower average relative affinities for the heterodimer sites relative to the monomers suggest (1) an artificial exaggeration in the differences in binding (in fact, in absolute terms the difference in monomer binding in ScrT1 >UbxT3 versus ScrT1 =UbxT3 is greater than the difference in heterodimer binding affinities for the same two groups) and (2) that the low affinity binding sites need to be compensated by relatively higher concentrations of the heterodimers, if they are relevant to gene expression. This should be discussed when the data are first reported on and dealt with in the discussion – see later comment.

We apologize for the confusion, which we think is because the labels in Figures 2d and 2e refer to relative affinity, not absolute affinity. To clarify, the average monomer relative affinity near Scr>Ubx peak center is about 0.05, and the average heterodimer relative affinity is 0.006. All relative affinities are normalized to the genome-wide maximum (defined as relative affinity of 1). As the reviewer suggests, many in vitro binding studies demonstrate that Hox-Exd heterodimers have a much higher absolute affinity compared to Hox monomers. As a result, one cannot compare the relative affinities for Scr and Scr-Exd. Nevertheless, based on previous Kd measurements on Hox monomers and Hox-Exd heterodimers, the Scr-Exd heterodimer indeed has a higher absolute affinity than Scr monomer.

5. Figure 4D top graphs do not have scale bars. The image looks like the scale bars are probably the same, but I'm betting that, based on Figure 2d and 2e, they are not.

Again, we apologize for not making this clear in the original submission. Two types of scales are used in this study. The scale for the *NRLB* scores has a clear biophysical meaning behind it: it indicates the relative affinity of a sequence, based on the *NRLB* models, which are in turn derived from the SELEX-seq data. On the other hand, the scale bar of ChIP results are arbitrary numbers that indicate relative signal intensity (# reads), but without an easily interpretable biophysical meaning. In our manuscript, the numerical values of the ChIP scale are in the range of 10^1 to 10^2 , while the scale bar of *NRLB* results is about 10^{-3} to 10^{-2} .

In Fig 4d of the original submission, in order to show these two different datasets side-by-side, we applied scaling factors to the *NRLB* results (the scaling factors for monomer and dimer *NRLB* results are, as expected, different), so the numerical values of the *NRLB* data bar are in the same range as those of ChIP data. Using scaling factors is a common practice when generating heatmaps, and does not alter the patterns these maps reveal, nor does it alter any conclusions we draw from these heatmaps. In fact, even when showing only ChIP results, scaling factors are often necessary, because ChIPs with different antibodies often result in scales with different absolute values.

In the revised manuscript, we have separated Fig 4d into three individual sections: ChIP-seq, monomer NRLB and dimer NRLB. Heatmaps in other figures are also changed in similar ways. In the revised heatmaps, the NRLB scales indicate relative affinity. These are described in the revised Methods section.

6. I was initially expecting the Exd-binding (YWPR) mutant of Scr to behave more like a null with respect to the number of sex comb teeth; this is explained later in the paper where we are reminded that Exd is only nuclear (and therefore functional) in the Hth-expressing (peripheral/future proximal) domain of the leg discs. Thus, loss of formation of Exd heterodimers would not be expected to affect sex combs since they form on a more distal leg segment. It would be helpful for the authors to say something about not expecting the sex comb number to be affected since that structure forms outside the Exd nuclear domain.

Agreed, thank you for pointing this out. We have modified the text accordingly (lines 205).

Also, I'm not sure the same is true regarding the suppression of scutellar bristles in T1. Are these Exd-independent structures? Don't they form in the Exd nuclear domain?

The Sp bristles are in Hth domain (Exd nuclear domain). A leg disc co-stained for Ac and Hth has been added as a new figure panel to make this point (Fig 3f).

7. In the reporter gene constructs, the exact mutations that were made were not shared (as far as I can find in the paper). It seems like the mutations were complete deletions of the Scr-Exd sites. If so, THIS IS NOT THE RIGHT EXPERIMENT. If the point of this paper is that the differential binding of Hox::Exd and not binding of the homodimers that makes a difference in which genes are differential expressed in the two leg discs, then they need to replace the Scr-Exd site with an Scr monomer site and ask happens to reporter gene expression.

The reviewer is mistaken; the mutations we made were 4 bp substitutions, not deletions. We know from prior studies that such substitutions (e.g. TGATTAAT to TGCCCCAT) strongly inactivate Hox-Exd dimer binding. These types of mutations are often used in the field to ensure that heterodimer binding is eliminated. Although the experiment suggested by reviewer 2, converting the Scr-Exd dimer motif to an Scr monomer motif, would be interesting, by completely inactivating the Scr-Exd motifs, the results are sufficient to demonstrate these selected CRMs are direct Scr-Exd targets, which was the purpose of these experiments. The fact that they are Scr-Exd targets, instead of Scr monomer targets, is supported from the ChIP results showing that the Scr YPWM mutant no longer binds to these CRMs, demonstrating that binding is YPWM (and Exd)-dependent.

We have clarified the text to make these points clear (line 301).

8. Likewise, if they are building a case that it's the partner proteins that make for specificity – they need to replace the Scr-Dll site with an Scr monomer site instead of deleting the site entirely. Here, again, it is not obvious what got deleted.

Although we agree that this would be an interesting experiment, it is unfortunately not practical at this time because from our analysis the Scr-Dll input is not limited to only one or a small number of well-defined binding sites. Instead, it appears that there are multiple homeodomain binding sites that can accommodate the Scr-Dll dimer. As a result, the best experiment we are able to do at this time is to delete them, completely removing the Scr-Dll input. The deletion was a 40 bp fragment near ChIP peak center (stated in the manuscript).

9. I would like some reconciliation between this study and what has been discovered in the past. Is there any example out there showing that Hox monomers can regulate gene expression on their own?

Yes, in the haltere, Sean Carroll's group (as well as others) demonstrated that some targets, particularly those regulated by Hox proteins in cells that do not express *hth*, are repressed in an Exd-independent manner (for example PMID: 15753212 and PMID: 12070087). Although we do not know if other cofactors are involved in these particular examples, this is an interesting possibility that we now mention in the discussion (line 473).

I think it has been nicely established that Hox::Exd heteromers can regulate gene expression. If the binding affinity for Hox::Exd sites is really about 8X lower than that for Hox monomers, how are the heterodimer binding sites relevant?

We again think the confusion here can be resolved by distinguishing relative affinity and absolute affinity. Please see our response to comment 4 above. Most importantly, the relative affinities for a Hox monomer cannot be directly compared to the relative affinities of a Hox-Exd heterodimer.

How do Hox proteins behave in the presence of Exd (with or without DNA) – do they preferentially form dimers?

Hox-Exd dimer formation is usually weak but detectable (depending on the assay) in the absence of DNA. The presence of DNA with a Hox-Exd binding site significantly enhances heterodimer formation.

10. It would also be helpful to correlate the binding data to the RNAseq data. Are the ScrT1=UbxT3 bound genes expressed to equal levels in the T1 versus T3 disc. Are the ScrT1>UbxT3 expressed to higher levels in the T1 disc? Do the sites that are Exd-dependent correlate better with gene expression? How about the Scr-Dll sites? They have the RNAseq data and they have the binding data; seems like a straightforward analysis. I realize that binding does not equal gene expression, but in the end, it's the differences in gene expression that matter and only relating binding to regulation for a few chosen reporter genes (and then

inadequately addressing the big story in the paper of monomer versus heterodimer binding) seems inadequate given the data they have at hand.

We have now included a new set of analyses (Extended Data Fig 5) that connect the transcriptome measurements with binding. These new results suggest that Scr bound CRMs could either activate or repress target gene transcription, which is consistent with our newly included CBP ChIP results. Interestingly, for ScrT1>UbxT3 binding events (but not ScrT1≈UbxT3 binding), Exd-dependent Scr target CRMs tend to be enhancers, not silencers.

Reviewer #3 (Remarks to the Author):

In the current study, the authors try to address a significant developmental biological question- How do Hox proteins which have similar DNA binding properties in-vitro initiate paralog-specific gene regulatory networks in-vivo? Using the T1 and T3 legs in *Drosophila* as models which are specified by the Hox proteins Scr and Ubx respectively, the authors try to understand the mechanisms which govern the differences between the two structures. They perform ChIP-sequencing for tagged Scr and Ubx proteins in T1 and T3 leg imaginal discs and identify close to 8% of binding loci which are paralog specific. Using ATAC seq experiments and a bioinformatics tool (NRLB), the authors arrive at the conclusion that paralog-specific binding events are a result of Scr-Exd interactions. However, ChIP-sequencing for a mutant version of Scr protein which cannot bind Exd indicate that only a handful of Scr bound loci dependent on Exd. The authors, however, further pursue the idea that Exd is significantly responsible for paralog specific binding of Scr in T1 and justify this idea using other bioinformatics, mechanistic and transgenic assays. Further, the authors identify Dll as a novel cofactor of Scr which helps in Scr binding in regions where Exd is not expressed. Taken together, the authors claim to identify paralog-specific binding loci for the Scr protein in the T1 leg discs and try to justify that binding with cofactors like Exd and Dll are critical for such paralog-specific binding.

We are pleased that the reviewer agrees that we are addressing a significant biological problem.

Conceptual Issues

1. RNA sequencing analysis for the T1 and T3 leg imaginal discs suggest that only a handful of genes are differentially expressed between the two discs, notable of which were Scr and Ubx themselves. Previous studies have suggested that Ubx regulates the expression of Scr in T3 legs. This could suggest two possibilities: 1) differences between the T1 and T3 legs might occur due to subtle changes in the gene regulatory network which might manifest from differences downstream of Hox targets and not at the level of Hox targets themselves.
- 2) differences between T1 and T3 legs might be due to events occurring during the pupal stage. The T1 and T3 leg imaginal discs, thus, do not seem to constitute a sound system to establish the causal relationship between paralog specific binding of Hox proteins to the differences in phenotype observed between the T1 and T3 legs.

The two possibilities raised by the reviewer are not mutually exclusive and we think that both are true. In particular, we agree that there are likely many subtle changes in the gene regulatory networks in the two leg discs. This is supported by the relatively similar, but not identical, RNA-seq profiles of the two leg discs. However, these differences must ultimately be a consequence of the two Hox proteins, and because Hox proteins are transcription factors, they most likely initiate differences in the network by binding and regulating CRMs. In addition, there are certainly many indirect effects downstream of the initial set of direct Hox target genes. Nevertheless, the premise that we begin with, and experimentally test, is that differences in the binding specificities between these two Hox proteins matter and, further, we ask how are these differences arise mechanistically.

We also agree with the reviewer that there is likely to be additional target gene regulation by these Hox proteins at later (and earlier) stages. However, this doesn't mean that analyzing differences at the L3 stage isn't relevant, a conclusion that is supported by our analysis of multiple enhancers, such as the one from *doublesex*, that show differences at the L3 stage. So although we cannot claim to have identified all of the differences between T1 and T3 legs, our choice to study the wandering larvae stage is justified by experimental results. Our main goal was also to identify the mechanistic principles, which are likely to hold true at other developmental stages. We have now discussed these issues in the revised Discussion.

Analytical and validity issues and suggested improvements

Figure 1 and 2:

The authors perform ChIP-sequencing analysis for Ubx and Scr in T3 and T1 leg imaginal discs respectively. They identify close to 8% of binding loci which is bound differentially by Scr but not by Ubx. Additionally, the paralog specific binding sites observed by the authors is largely at intergenic regions.

There are a few issues with the results and interpretations here:

1. The term binding events, sites, loci and targets seem to be used very casually and are not consistent. This needs to be taken care of.

Agreed. We are now more precise and consistent in the revised manuscript.

2. Even though the authors do not provide a list of the targets that correspond to the differentially bound loci, they claim that there is a strong asymmetry between paralog specific targets of Scr and Ubx in T1 and T3.

This claim is based on the different numbers of differential binding sites (Fig 2a), which are objectively determined by software from the raw data.

Additionally, it needs to be investigated whether such paralog-specific binding events result in any differential gene expression between the T1 and T3 discs. Thus, claiming that differences in binding of the Hox proteins translates to gene regulatory changes seems premature.

In the initial submission, we addressed this issue by characterizing dozens of reporter genes that were derived from paralog-specific binding events in our CHIP experiments. We observed about one third of these display differential expression between T1 and T3 leg discs. These results provide clear functional links between paralog-specific binding and differential gene expression. Notably, the absence of differences seen for some CRMs may be due to the issue raised above by the reviewer, namely, that they may be active at later stages of development. Consequently, the one-third estimate may be an underestimate.

In the revised manuscript, we also include CHIP data for CBP, which is a marker for CRM activity, and provide a global overview of CRM binding. These data provide independent evidence that paralog-specific Hox binding is associated with active CRMs, and is not simply non-specific binding. Finally, additional analysis of the RNA-seq data, showing that paralog-specific Hox binding is often near genes that are differentially expressed in the T1 and T3 leg discs, also support this conclusion.

3. The authors indicate that most paralog-specific binding events occur at the intergenic regions and that CRMs are rich in paralog-specific loci. While the authors do explain in detail the methodology of their assay, a mere 8% paralog specific binding loci (most of which are in intergenic regions) can easily be attributed to non-specific binding, as is usually the case in all genome wide studies. The authors need to be careful here while making such claims and should provide sufficient data to back their statements.

We agree that it is challenging to determine if a particular binding event is functional, and most papers in the literature do not attempt to do this. However, it is incorrect to suggest that the paralog-specific binding events we identified are non-specific, based solely on their percentage. Such a percentage reflects the biological nature of the particular case being studied (e.g. Hox binding in T1 and T3 legs), and should not be subjectively judged as being high enough to be valid, or too low and therefore non-specific.

As described above, we now have several lines of evidence that suggest at least some of the paralog-specific binding is functional: 1. Reporter assays demonstrate that many of the Scr>Ubx bound loci have CRM activity. 2. The ScrT1>UbxT3 loci have a clear Hox-Exd binding signature, based on both bioinformatic analysis, as well as comparison between wild type and YPWM mutated Scr CHIP results. 3. CBP is bound to the same CRMs that have paralog-specific Hox binding. 4. Transcriptome differences between the two leg discs are observed in genes near paralog-specific binding events.

4. De-novo motif analysis of peaks from Scr and Ubx CHIP-seq suggest enrichment of motifs which the authors describe to be Hox-Exd motifs. However, later analysis (see point 7, point 8)

indicate that majority of the Scr bound sites are Exd independent. This seems to be contradictory and needs to be properly addressed.

There is no contradiction. The motif discovery methods compare a custom defined set of loci (the test set, in this case all Scr binding loci) with a set of randomly selected loci from the genome. If a particular motif happens rarely in the random set, it does not need to be present in the majority of the test set to be identified as a statistically significant enriched motif. In the case of Scr, the dimer motif we report in Figure 1h is determined by the software as highly significant (p value of 10^{-7}).

The authors should, additionally, make sure that what they claim to be as Hox-Exd sites based on SELEX studies are actually Hox-Exd sites in-vivo also using proper experimental evidence.

In this comment, the reviewer is implying that the Hox-Exd sites determined by in vitro SELEX experiments may not reflect Hox-Exd sites in vivo. Based on many years of studying these binding events in vitro and in vivo, by us and other labs, we are very confident that it is not the case. We base the determination of Hox-Exd sites in vivo on many publications from our lab (for example PMID: 10398683 and PMID: 20634319) as well as by other labs (for example PMID: 8657149, PMID: 28784834, PMID: 18417536, PMID: 12923056 and PMID: 9010234). In many cases, both in this paper and in previous publications (by us and by others), we have confirmed that the Hox-Exd site identified by sequence and binding is in fact required for enhancer activity in vivo, using reporter genes and in vitro DNA binding measurements.

5. The fact that binding sites for Ubx in T3 are much lower than Scr in T1 is very interesting. Previous studies have shown that development of the T3 mesothorax is achieved in part by repression of Scr by Ubx. Analyzing whether the paralog specific binding of Ubx in T3 is at the Scr loci can interesting clues to mechanisms governing differences in developmental programs between T1 and T3.

We have provided evidence for cross-regulation between Scr and Ubx below. We agree that these are interesting observations, but further analysis of the downstream networks is beyond the scope of this manuscript.

6. In Line 156-158, the authors report that both difference in gene expression as well as chromatin accessibility were found to be different for the Scr and Ubx loci. This would somewhat indicate that regulation of one Hox protein by the other is a major determinant of T1 and T3 leg specification. The minor differences in gene regulatory networks between T1 and T3 can be due to events downstream of Hox direct targets or during the pupariation rather than a direct consequence of paralog-specific binding. To counter this argument, as already suggested, the authors should provide a causal relationship of paralog-specific binding to the difference in the T1 and T3 leg phenotypes.

As stated in our response above, we are not arguing against cross-regulation between Scr and Ubx, or the contributions of indirect Hox targets and other developmental stages. On the other

hand, these issues are beyond the scope of our current study and are not particularly relevant to the conclusions of this work.

We have included results of dozens of reporters from selected paralog-specific binding targets, and showed their expressions in leg imaginal discs. For many, we show that the binding site is essential for expression, thus providing a causal link between binding and activity.

In addition, in the revised manuscript, we also included CBP binding data, which further links paralog specific binding events to CRM activity.

7. The authors next turn to NRLB model to identify whether monomeric Hox binding motifs or Hox-Exd motifs are enriched in paralog-specific loci. In Line 170-174, the authors observe a similar enrichment score for Scr and Ubx monomeric binding at the ScrT1≈UbxT3 loci. Interestingly, they find a similar result for ScrT1>UbxT3 loci. They also find that relative enrichment of Scr-Exd motif is much greater ScrT1>UbxT3 loci as compared to ScrT1≈UbxT3 loci, thereby inferring that Exd might have an important role to play in identifying paralog specific targets.

There are two issues with the results here. First, it looks like there is disparity between de-novo search algorithms (which finds Hox-Exd motif to be the enriched motif) and the NRLB models (which finds monomeric Hox motifs to be enriched at the peaks).

The “discrepancy” the reviewer refers to can be explained by the different emphasis of these analyses. In motif search, the algorithm mainly determines if a motif is “significantly enriched” in the test set compared to a random set. In the *NRLB* analysis, the software looks if a motif is present, and how strong the motif is based its computed relative affinity.

Although the Scr-Exd motif is present in only about 10% of all Scr peaks, it was determined to be significantly enriched compared to the background, as determined by commonly used software in the field. Although It is possible that the Scr monomer motif is present in a higher percentage of all Scr peaks (for example 20%), it is present in a comparable percentage of the background sequences, due to the degenerate nature of these binding sites, and therefore deemed not statistically significant.

Second, authors completely ignore the fact that the relative enrichment of monomeric binding sites at ScrT1≈UbxT3 loci as well as ScrT1>UbxT3 loci doesn't necessarily mean that they cannot impart specificity of binding.

If the reviewer is suggesting that monomeric Hox binding sites might impart specificity of binding, we cannot agree because our data clearly argue the opposite: Hox monomer binding

sites alone are not sufficient to lead to paralog-specific binding *in vivo*, and we have made this point clear in the manuscript (line 182). If Hox monomer binding was sufficient, in Scr_{T1}>Ubx_{T3} loci, we would expect higher average relative affinity of Scr monomer motifs than that of Ubx monomer motifs, which we did not observe (the two averages are in fact roughly equal). The scoring of monomer binding affinities is based on the analysis of SELEX-seq data with *NRLB*, which uses tens of millions of DNA sequences to accurately compute relative affinities.

If the reviewer is suggesting that Scr_{T1}≈Ubx_{T3} loci might lead to T1≠T3 target gene transcription, we agree with this point. In fact, our RNA-seq results (Ext. Data. Fig 5) support this notion. However, although there may be mechanisms that result in differential gene expression that are independent of DNA binding specificity, this is not the focus of our current study, and we are more than happy to systematically address this issue in subsequent studies.

The expression of the Hox proteins and their relative dosage are other factors which can influence Hox specificity of binding to these loci.

Thus, the statement that Exd significantly contributes to paralog-specific Hox binding, at best, seems premature. The authors do not show substantial validations to support such a claim.

We disagree. The statement that Exd significantly contributes to paralog-specific Hox binding is based on data showing that many Scr>Ubx binding events are no longer observed in the YPWM mutant ChIPs (Fig 5a). As Exd is the only known protein to bind to the YPWM motif, these data strongly support the notion that these sites are Exd-dependent. Moreover, they are also supported by the analysis of these (and other) peaks using our NRLB models: the sites that are dependent on the YPWM motif have a clear Hox-Exd binding signature. Finally, in the revised manuscript, we show that these sites also bind Hth, which is a surrogate for Exd binding, as described above.

The question of whether the dosage of Hox proteins has any effects on paralog-specific binding is interesting but is not the focus of this study.

Fig 4 and Fig 5:

8. The authors next move on to generate a Scr mutant (Scr (YPWM*)T1) that is unable to interact with Exd to test if heterodimerization with Exd contributes to paralog specific binding. Interestingly, the authors here compare the ChIP-seq for the wild type and mutant Scr and report that majority of sites are Exd independent. De-novo motif analysis for peaks in Scr (YPWM*)T1 seems to indicate that monomeric Scr binding site is the preferred motif. There are a few inconsistencies and issues with the finding here. First, the very fact that de-novo motif analysis for the wild type Scr shows enrichment of what is termed as the Scr-Exd motif, even when most sites are Exd independent, seems very contradictory.

As described above, motif discovery just means there is a significant enrichment, not that all peaks have the motif.

Second, while the authors try to support their finding saying that in the Scr (YPWM*)T1 mutant, the Scr-Exd motif is not enriched, the data in Extended figure 3b is hardly encouraging. The authors seem to ignore the first motif which is very similar to what they term as the Scr-Exd motif and is highly enriched. While the authors should take negative results into account and arrive at their conclusions based on what is observed, I feel it is time to redefine what is suggested as Hox-Exd motif and what as Hox monomeric motif based on experimental data like ChIP-seq and not simply based on in-vitro Selex seq studies. The authors should not make flamboyant conclusions (like the one made in Line 226-228) while not properly taking their results into consideration as well as not providing enough experimental evidence to back their claims.

This is a harsh comment that appears to be based on a misinterpretation of what we are saying and the many publications on this topic already in the literature. Based on a wealth of experimental evidence, not just SELEX-seq, we can confidently say which motifs are Hox-Exd and which are not. In summary, we are not ignoring negative results, but are basing our conclusions on solid data and analysis.

We also take issue with the reviewer's claim that our statements are "flamboyant" and their suggestion that we are ignoring negative results. It appears that this reviewer made these claims solely based on his/her subjective judgment by eye that the first Scr(YPWM*) motif in Extended Fig 3b looks similar to an Scr-Exd motif. This is not the case, as we justify in detail below.

The consensus Scr-Exd motif is TGATTAAT (for example PMID: 22153072 and PMID: 29610332). For convenience, we will compare the reverse complement logos of the first motif enriched in wild type and YPWM mutated Scr ChIP peaks (see Extended Data Fig 3b). The former represents a well defined Scr-Exd motif, while the latter is not an Scr-Exd motif because it is missing multiple critical elements.

For example, we know that a G at position 2 is highly favored. Although G is not the top base at position 2 of the wild type motif, it is close. On the other hand, G ranks last in the mutant motif, which would make this motif a very unfavorable binding site for Scr-Exd dimer.

We also know from previous work that an AT dinucleotide at positions 7 and 8 is critical for the binding of the Scr homeodomain. The wild type motif clearly has this dinucleotide at the right position, but the mutant motif does not. Moreover, the mutant motif has an AT dinucleotide at positions 6 and 7, which disrupts the necessary spacing between the Exd half site (TGAT) and the Scr half site (TAAT). Such a conformation is highly disfavored by the Scr-Exd heterodimer.

Finally, as an objective test, we replaced the TGATTAAT core Scr-Exd motif from the *h-1* CRM we reported in our manuscript, with the top sequence in the mutant motif above (TCATGATA), and used the *NRLB* algorithm to quantify the Scr-Exd relative affinity. This replacement changed the Scr-Exd relative affinity of this locus from 0.19 (a high affinity site) to $<10^{-7}$ (essentially non-specific binding).

9. The authors next move onto confirm the importance of Scr-Exd interactions on paralog specific binding events. In line number 237-238, the authors suggest that of the 432 ScrT1>UbxT3 loci, 183 sites seem to be Exd dependent as suggested from ScrT1/Scr(YPWM*)T1 ChIP-signal ratio. Later they do the same analysis for ScrT1≈UbxT3 peaks.

This is where it gets really confusing as the authors suggest that 1545 of 5346 peaks have ScrT1/Scr(YPWM*)T1 ChIP-signal ratio greater than 2. However, previous comparison between ChIP-seq for ScrT1 and Scr(YPWM*)T1 suggested that only 667 binding sites were found to be Exd dependent. This data does not reconcile.

Here the reviewer appears to be confused by the following numbers we reported, which we acknowledge was a problem that we have now corrected. Previously, we identified 667 Exd-dependent peaks from all Scr peaks in Fig 4d, but from ScrT1≈UbxT3 peaks, which should be a subset of all Scr peaks, more peaks (1545) were reported as Exd-dependent (Fig 5b in the original submission). This difference was due to using different criteria for Exd-dependency in these two figure panels, and we apologize for not explaining this more clearly. In Fig 4d, as stated in methods, we treated peaks with $FDR < 0.05$ (reported by DiffBind) as Exd-dependent, while in Fig 5b (as well as Fig 5a) of the original submission, as described in the text, we defined peaks with ScrT1/Scr(YPWM*)T1 ChIP ratio > 2 as Exd-dependent.

While our analyses in these figures were correct, to avoid any confusion, in the revised manuscript, we use only FDR as the criteria to determine Exd-dependency. We repeated all analyses that were based on the ChIP ratio criteria with the FDR criteria, and reached the same conclusions as were in the original manuscript.

Also, in Figure 5b, the heatmaps are plotted for 2152 peaks? There is no mention of where they got this number from. The authors should make sure that their data is consistent in the text as well as on the images.

These heat maps plotted a subset of Scr~Ubx sites that are located in the intron or intergenic regions (biased towards CRMs for the reasons stated above).

Because the ScrT1~UbxT3 loci are not the focus of this study, we decided to remove this figure panel from the revised manuscript to keep it more focused on paralog-specific binding. This change does not affect the main conclusions of our study.

Figure 6

10. The authors then go on to generate lacZ reporters to reconfirm the paralog specific activity of Scr and its dependence on Exd using a transgenic approach. While their results do indicate that a fraction of Scr paralog specific loci drive T1>T3 expression, the authors do not provide substantial evidence to claim that such differences are due to the direct regulation by Scr-Exd complex. To identify whether Exd has a role here, the authors should perform genetic assays (perhaps by making Exd clones or showing Exd occupancy using ChIP-seq) rather than just mutating the Scr-Exd sites, as it is somewhat evident from the rest of the data that monomeric Scr might also have a significant role to play in paralog-specific target selection via motifs that, although are referred to as Scr-Exd sites, might as well be high affinity binding motifs for the Scr protein itself.

This reviewer has acknowledged here that our results “do indicate that a fraction of Scr paralog specific loci drive T1>T3 expression”. This apparently contradicts earlier statements that we did not establish connections between paralog-specific binding and differential expression between T1 and T3 leg discs. Such contradictions in the comments are confusing, but we have done our best to explain and improve the presentation of the data.

We agree that it is critical to establish direct regulation between Scr-Exd and its target CRMs. We however disagree with the approaches suggested by the reviewer here. The approach we took in our study, making 4 bp mutations that specifically eliminated Scr-Exd sites, is the best way to determine if Scr-Exd directly regulates the CRM being tested. Indeed in several cases, these mutations abolished reporter expression, which provides strong evidence that Scr-Exd directly regulates these CRMs.

On the other hand, making Exd clones, as the reviewer suggested, can only put Scr or Exd genetically upstream of the candidate CRM, but cannot distinguish between direct vs. indirect regulation. Exd ChIP provides information on co-occupancy at the CRMs but does not reveal functional regulation of the CRMs. We also now include Hth ChIP data in the revised manuscript as a readout of genome-wide Exd occupancy. We show that the Hth ChIP results agree with our other results.

REVIEWERS' COMMENTS

Reviewer #2 (Remarks to the Author):

I am very pleased with this modified manuscript demonstrating the role of Hox cofactors in the differential expression of genes in homologous structures. All of my concerns were addressed either by changes in the text and figures or in the response to reviewers. The addition of the Hth-ChIP-Seq data solidifies the characterization of changes in Scr binding with Scr (YPWM) mutant as being a result of the loss of Exd cofactor binding.

I would recommend one change. The label on panel 1d is a bit confusing - I expected to see that T1-Specific genes would be shown as positive fold change, not the T3-specific genes. Either re-label or show the data differently in a log₂T1 versus log₂T3.

Reviewer #3 (Remarks to the Author):

I am satisfied with the revision in response to the comments on the first version of the MS. I appreciate authors taking time and efforts to generate additional evidence and to re-write the MS.

REVIEWERS' COMMENTS

Reviewer #2 (Remarks to the Author):

I am very pleased with this modified manuscript demonstrating the role of Hox cofactors in the differential expression of genes in homologous structures. All of my concerns were addressed either by changes in the text and figures or in the response to reviewers. The addition of the Hth-ChIP-Seq data solidifies the characterization of changes in Scr binding with Scr (YPWM) mutant as being a result of the loss of Exd cofactor binding.

I would recommend one change. The label on panel 1d is a bit confusing - I expected to see that T1-Specific genes would be shown as positive fold change, not the T3-specific genes. Either re-label or show the data differently in a $\log_2 T1$ versus $\log_2 T3$.

Thank you for the positive feedback; we have made the suggested change to Figure 1d. For clarity, we added the labels T3>T1 to the top half of the graph and T1>T3 to the bottom half of the graph.

Reviewer #3 (Remarks to the Author):

I am satisfied with the revision in response to the comments on the first version of the MS. I appreciate authors taking time and efforts to generate additional evidence and to re-write the MS.

Thank you for the positive comments!